# GLOBAL OPTIMALITY OF IN-CONTEXT MARKOVIAN DYNAMICS LEARNING

## ABSTRACT

Transformers have demonstrated impressive capability of in-context learning (ICL): given a sequence of input-output pairs of an unseen task, a trained transformer can make reasonable predictions on query inputs, without fine-tuning its parameters. However, existing studies on ICL have mainly focused on linear regression tasks, often with i.i.d. inputs within a prompt. This paper seeks to unveil the mechanism of ICL for next-token prediction for Markov chains, focusing on the transformer architecture with linear self-attention (LSA). More specifically, we derive and interpret the global optimum of the ICL loss landscape: (1) We provide the closed-form expression of the global minimizer for single-layer LSA trained over random instances of length-2 in-context Markov chains, showing the Markovian data distribution necessitates a denser global minimum structure compared to ICL for linear tasks. (2) We establish tight bounds for the global minimum of single-layer LSA trained on arbitrary-length Markov chains. (3) Finally, we prove that multilayer LSA, with parameterization mirroring the global minimizer's structure, performs preconditioned gradient descent for a multi-objective optimization problem over the in-context samples, balancing a squared loss with multiple linear objectives. We numerically explore ICL for Markov chains using both simplified transformers and GPT-2-based multilayer nonlinear transformers.

## 1 INTRODUCTION

Transformer-based large language models (LLM) have demonstrated advanced capability of in-context learning (ICL): given a prompt, consisting of input-label pairs, a trained transformer can predict the label for an unseen input without updating its parameters (Brown, 2020; Rae et al., 2021; Garg et al., 2022; Liu et al., 2023; Team et al., 2023; Achiam et al., 2023; Touvron et al., 2023). This ability to solve novel tasks solely from examples not only provide a potential alternative for expensive fine-tuning (Li et al., 2024b), but also enhance reasoning tasks like chain-of-thought (Lampinen et al., 2022), self-correction (Wang et al., 2024), with applications in mathematical problems and logical deduction (Wei et al., 2022).

The ability of transformers to solve unseen tasks in-context has sparked a line of research investigating the underlying mechanisms from various perspectives, including expressive power (Von Oswald et al., 2023; Akyürek et al., 2023; Giannou et al., 2023; Li et al., 2023; Dai et al., 2023; Zhao et al., 2023; Bai et al., 2024), convergence of transformer training dynamics (Zhang et al., 2024; Huang et al., 2023), generalization ability (Duraisamy, 2024; Li et al., 2023; 2024a), and optimization theory and global optimality (Ahn et al., 2023; Mahankali et al., 2024; Li et al., 2024b). In particular, (Ahn et al., 2023) identified a distinctive sparse structure in the global optimal transformer parameters, by setting some entries of the model parameters directly to zero, which simplifies the structure of the solution. Building on this sparsity, they demonstrated that the forward pass of linear attention models implements preconditioned gradient descent.

However, the tasks considered in prior studies are limited to linear regression or classification, where both feature and task vectors are zero-mean Gaussian, which offers limited insight into how transformers learn sequential data governed by specific dynamics in-context. For example, when presented with examples of math word problems that include intermediate steps and answers, an LLM can generate reasonable answers to new questions (Lampinen et al., 2022). Nevertheless, the relationships among these examples cannot be directly modeled using linear functions with

Gaussian-distributed data. Instead, they resemble sequences governed by dynamic processes over a vocabulary, which can be conceptualized as a discrete state space. Therefore, investigating how transformers learn such dynamics-based data in-context is essential to building a more systematic understanding of ICL. In particular, we focus on the ICL for Markov chains, a classic model used to represent language (Shannon, 1948; 1951; Makkuva et al., 2024).

**Major challenges.** The challenges posed by in-context Markovian dynamics learning are two-fold: (i) The objective function is non-convex w.r.t. transformer parameters, due to their nonlinear coupling, which complicates the identification of the global minimum. To mitigate this, we transform the problem through reparameterization to a strictly convex optimization that produces either the global minimum or a tight lower bound, inspired by Ahn et al. (2023). (ii) Since the next token is stochastically dependent on the previous tokens, no analytic expression exists for the labels in the ICL setting. This introduces an additional layer of randomness beyond the feature and task vectors. Specifically, compared to the linear case, we also need to consider the randomness of the label conditioned on the feature and task vectors.

**Our contributions.** To this end, we study how transformers learn to predict the next token for Markov chains in context by analyzing the loss landscape of linear self-attention (LSA) models. Given the challenges posed by non-convexity and stochasticity, we focus on binary Markov chains with first-order memory as our first step. The major contributions of this work are highlighted as follows.

▶ We establish a framework for handling ICL with Markovian dynamics by fully characterizing the global minima of the loss landscape for the LSA model trained on length-2 binary Markov chains. This analysis applies to both i.i.d. settings (see Proposition 1) and general initial-state distributions (see Proposition 2). Our results show that the global optimum adapts to the Markovian dynamics, exhibiting a denser structure compared to ICL for linear regression. In comparison to traditional i.i.d. tasks, additional nonzero model parameters in the Transformer are necessarily included for achieving the global minimum of the loss due to the temporal dependence within the in-context samples.

▶ To the best of our knowledge, our theoretical result is the first to provide a closed-form expression for the lower bounds of the expected global optimal value in next-token prediction using a one-layer transformer structure for Markovian data of arbitrary length. Building on this result, we further derive an upper bound by properly selecting the transformer parameters.

▶ We advance the understanding of multilayer transformer expressivity by exploring a parameter subspace that mirrors the structure of the derived global minimum for Markovian dynamics. Our results show that the forward pass of the multilayer linear transformers is equivalent to solving a multi-objective optimization problem. This problem minimizes a squared loss while simultaneously maximizing multiple linear objectives (see Proposition 3).

**Related work.** The capability of transformers to perform ICL (Brown, 2020; Rae et al., 2021; Liu et al., 2023; Garg et al., 2022) has inspired an exploration of its underlying mechanism from various aspects. A line of works have shown transformers trained on in-context prompts implicitly implement optimization algorithms. Akyürek et al. (2023) constructed a set of weights in transformers such that their forward pass is equivalent to a step of gradient descent over the mean squared loss on in-context examples. Von Oswald et al. (2023) provided such a construction for LSA, further showing actual optimization of transformers on in-context loss landscapes converge to such a construction. In addition to standard learning algorithms such as least squares and ridge regression, Bai et al. (2024) showed that transformers implement algorithm selection. Specifically, transformers first determine the task type based on the data statistics in the prompt and then choose the most optimal standard algorithm to make predictions for the query input.

From the perspective of optimization theory, Mahankali et al. (2024); Ahn et al. (2023) showed trained LSA networks emulate preconditioned gradient descent via analyzing the loss landscape. Gatmiry et al. (2024) proved that the global minimizer implements multi-step preconditioned gradient descent, considering looped transformers (Giannou et al., 2023). While previous works mainly focused on i.i.d. in-context examples, Li et al. (2024b) further analyzed the ICL loss landscape under correlated designs, in addition to the consideration of state-space model and LoRA. There has also been studies about the training dynamics of transformers in the ICL setting. Zhang et al. (2024) demonstrated that LSA trained through gradient flow converges to the global minimum under mild distribution shifts, achieving close performance to the best linear predictor. Huang et al. (2023) proved convergence

of training dynamics to near-zero prediction error under both balanced and unbalanced in-context samples. Another relevant area of our work is time-series prediction, which we discuss in section F. The comparison between this work and existing research is summarized in Table 1.

A line of concurrent work has studied transformers for temporal data structures, including Markov chains Makkuva et al. (2024); Sander et al. (2024); Rajaraman et al. (2024); Nichani et al. (2024). These studies primarily focus on attention mechanisms operating within a single Markov chain. In contrast, our work takes a complementary approach by examining a controlled setup where transformers learn the similarities between entire sequences rather than within individual Markov chains. This perspective enables us to explore how transformers manage complex dependencies across sequences, particularly in settings with non-Gaussian input distributions and non-linear input-output relationships. Notably, this work, to the best of our knowledge, represents the first step toward understanding the attention mechanisms involved in extracting sentence-level relationships between prompts and queries. This serves as a complementary contribution to characterizing the expressiveness of Transformers for Markovian data.

Table 1: Comparison with existing works on transformers for Markov chains.

| Work | ICL | Data | Non-i.i.d. In-Context Input | Optimum w/ Attention |
|------|-----|------|------------------------------|----------------------|
| Zhang et al. (2024) | ✓ | Gaussian | | |
| Mahankali et al. (2024) | ✓ | Gaussian | | ✓ |
| Ahn et al. (2023) | ✓ | Gaussian | | |
| Li et al. (2024b) | ✓ | Gaussian | ✓ | ✓ |
| Makkuva et al. (2024) | ✓ | Markovian | N/A | |
| Nichani et al. (2024) | | Causal | ✓ | |
| Rajaraman et al. (2024) | ✓ | Markovian | ✓ | |
| Sander et al. (2024) | ✓ | Autoregressive | ✓ | ✓ |
| Ours | ✓ | Markovian | ✓ | ✓ |

**Organization of this paper.** The paper is organized as follows. In section 2, we introduce the preliminaries, including data distribution, architecture, and the training objective. Our main theoretical findings regarding global optimality and expressivity are presented and validated in section 3. Finally, we conduct experiments on multilayer GPT-2-based transformers trained on in-context Markovian data in section 4, demonstrating improved accuracy compared to LSA and baseline learning algorithms, such as logistic regression.

## 2 PRELIMINARIES

### 2.1 IN-CONTEXT LEARNING

ICL refers to the operation on a prompt consisting of $n$ input-output pairs and a query input:

$$\boldsymbol{p} = (x_1, y_1, \ldots, x_n, y_n, x_{n+1}) = (\{(x_i, y_i)\}_{i=1}^n, x_{n+1}) \tag{1}$$

where $y_i = h(x_i)$, $\forall i \in [n+1]$ for some unknown function $h \in \mathcal{H}$, and $x_i, y_i$ belong to some input space $\mathcal{X}$ and output space $\mathcal{Y}$, respectively. ICL aims to form an output $\hat{y}_{n+1}$ for the query input $x_{n+1}$ that approximates its true label $\hat{y}_{n+1} \approx h(x_{n+1})$. The function $h : \mathcal{X} \to \mathcal{Y}$ remains the same within a single prompt yet varies across prompts.

Prior works have focused on linear function space $\mathcal{H}$: $h(x) = y = w^\top x$ for some $w \in \mathcal{X}$. Under such a construction, $y$ is deterministic once $x$ is provided. Despite being commonly encountered in many real-world applications, the case where $h$ is stochastic remains largely unexplored. For example, $h$ can represent a text generation mechanism that provides descriptions revolving a given topic. Then the token generated in the next step is associated with a probability based on the previously generated words Chorowski & Jaitly (2016). To approach the ICL for such scenarios, we consider a simplified

setting of next token prediction for Markov chains. The state space resembles vocabulary and the transition probability is akin to the conditional probability of the next word given the previous text.

## 2.2 MARKOV CHAINS

The evolution of a Markov chain $s$ of order $k$ on a state space $\mathcal{S}$ depends solely on the $k$ most recent states. For time step $\tau \in \mathbb{Z}_{\geq 1}$, we let $s_\tau$ denote $\tau$th state in the sequence $s$, the probability of observing state $j \in \mathcal{S}$ at time step $\tau + 1$ is:

$$\mathbb{P}(s_{\tau+1} = j \mid s_{1:\tau}) = \mathbb{P}(s_{\tau+1} = j \mid s_{\tau-k+1:\tau}) \tag{2}$$

where $s_{\tau_1:\tau_2}$ denotes the subsequence from time step $\tau_1$ to $\tau_2$. For first-order Markov chains, the dynamics are determined by the transition probabilities $p_{ij} := \mathbb{P}(s_{\tau+1} = j \mid s_\tau = i)$, which indicate the probability of transitioning from state $i \in \mathcal{S}$ to state $j \in \mathcal{S}$. These probabilities constitute the Markov kernel $\mathsf{P} = (p_{ij}) \in [0,1]^{|\mathcal{S}| \times |\mathcal{S}|}$. For a binary state space $\mathcal{S} = \{0, 1\}$, The transition matrix for a binary Markov chain is represented as $\mathsf{P}(p_{01}, p_{10}) := [1 - p_{01}, \ p_{01} \ ; \ p_{10}, \ 1 - p_{10}]$. Let $\pi_\tau \in [0,1]^{|\mathcal{S}|}$ denote the marginal probability at the $\tau$th time step. The relationship between consecutive time steps is given by $\pi_{\tau+1} = \pi_\tau \mathsf{P}$. A binary Markov chain $s \sim (\pi_1, \mathsf{P}(p_{01}, p_{10}))$ can be generated by starting with an initial distribution $\pi_1$ and iteratively applying $\mathsf{P}(p_{01}, p_{10})$ to update the state probabilities at each time step.

## 2.3 DATA FORMALISM

We introduce the input embedding matrix formulation used for our theoretical results. For a Markov chain $s$ with length $d + 1$, we take its first $d$ states to be the input $x = s_{1:d}$ and the final state to be the output $y = s_t$. The input and output space are $\mathcal{X} = \mathcal{S}^d$ and $\mathcal{Y} = \mathcal{S}$. We use subscripts to denote the indices of in-context samples, such that $x_i$ represents the first $d$ time steps of the $i$th in-context Markov chain, while $y_i$ denotes its final state. To form an input embedding matrix $Z_0 \in \mathbb{R}^{(d+1) \times (n+1)}$, we stack $(x_i, y_i) \in \mathbb{R}^{d+1}$ as the first $n$ columns and let the last column be $(x_{n+1}, 0)$, inspired by Zhang et al. (2024).

$$Z_0 = \begin{bmatrix} z_1 & z_2 & \cdots & z_n & z_{n+1} \end{bmatrix} = \begin{bmatrix} x_1 & x_2 & \cdots & x_n & x_{n+1} \\ y_1 & y_2 & \cdots & y_n & 0 \end{bmatrix} \tag{3}$$

where $z_i \sim (\pi_1, \mathsf{P}(p_{01}, p_{10}))$ for initial probability mass function $\pi_1 = [1 - p, p]$ with $p \in (0, 1)$ and transition probabilities $p_{01}, p_{10} \sim U(0, 1)$. The Markov kernel varies for each prompt, while the initial probability $p$ remains constant across all prompts. Let TF denote a transformer-based autoregressive model. The goal of ICL is to learn a model TF that can accurately predict the label of the query input:

$$\hat{y}_{n+1} := \mathtt{TF}(Z_0) \approx y_{n+1} \tag{4}$$

## 2.4 MODEL AND TRAINING OBJECTIVE.

We consider transformers with LSA layers (Von Oswald et al., 2023; Zhang et al., 2024; Ahn et al., 2023; Schlag et al., 2021). We recall a single-head self-attention layer (Vaswani et al., 2017) parameterized by key, queue, value weight matrices is defined as follows:

$$\mathrm{Attn}_{W_{k,q,v}}(Z) = W_v Z M \cdot \mathrm{softmax}\left(Z^\top W_k^\top W_q Z\right), \quad M := \begin{bmatrix} I_{n \times n} & 0 \\ 0 & 0 \end{bmatrix} \in \mathbb{R}^{(n+1) \times (n+1)} \tag{5}$$

where $W_k, W_q, W_v \in \mathbb{R}^{(d+1) \times (d+1)}$ are the (key, queue, value) weight matrices and $I_{n \times n}$ denotes the identity matrix. The attention scores are normalized by the softmax operator. The mask matrix $M$ reflects the asymmetric prompt due to the absence of the label for $x^{(n+1)}$. Motivated by Ahn et al. (2023); Zhang et al. (2024), we simplify the architecture by (i) removing the softmax nonlinearity and (ii) reorganizing the weights as $P := W_v$ and $Q := W_k^\top W_q$, merging the query and key matrices into a single matrix:

$$\mathrm{Attn}_{P,Q}^{(\mathrm{lin})}(Z) = PZM(Z^\top QZ). \tag{6}$$

Despite its simplicity, LSA demonstrates ICL capability for linear functions (Zhang et al., 2024) and has been shown to implement gradient descent (Von Oswald et al., 2023) and preconditioned gradient

descent (Ahn et al., 2023) to solve linear regression in-context. We will prove in section 3.2 that certain parameter configuration implements preconditioned gradient descent for a multi-objective optimization problem that includes linear regression. Finally, our architecture consists of $L$-layer LSA modules. Let $Z_l$ denote the output of the $l$th layer attention, we have

$$Z_{l+1} = Z_l + \frac{1}{n} P_l Z M(Z^\top Q_l Z) = Z_l + \frac{1}{n} \text{Attn}_{P_l,Q_l}^{(\text{lin})}(Z_l) \quad \text{for } l = 0, \dots, L-1. \tag{7}$$

The normalizing factor $n$ averages the attention weights gathered from the in-context examples. We consider the output of the transformer to be the bottom-right entry of the $L$th layer, i.e., $\text{TF}_L(Z_0; \{P_l, Q_l\}_{l=0,\dots,L-1}) = [Z_L]_{(d+1),(n+1)}$. To train the in-context learner, we optimize the following population loss in the limit of an infinite number of training prompts such that each prompt corresponds to a distinct Markov kernel $\{p_{ij}\}_{i,j\in\mathcal{S}}$:

$$f(\{P_l, Q_l\}_{l=0,\dots,L-1}) = \mathbb{E}_{Z_0,\{p_{ij}\}_{i,j\in\mathcal{S}}}[\ell(\text{TF}_L(Z_0; \{P_l, Q_l\}), y_{n+1})] \tag{8}$$

where $\ell(\cdot, \cdot)$ is the point-wise error. In the following section, we primarily focus on the square loss and provide a brief discussion of the global minimum in the case where $\ell$ is the cross-entropy loss. Our data distribution, architecture, and main finding can be summarized in Fig. 1.

Figure 1: Comparison between the sequence-level in-context Markovian data based attention structures and the existing works. (a) The key difference is that the exiting studies of the attention mechanism (Makkuva et al., 2024; Sander et al., 2024; Rajaraman et al., 2024; Nichani et al., 2024) is adopted on a token-level, whereas our study studies sequence-level attention. (b) While prior work samples in-context input and task vectors independently from some given Gaussian distribution (Ahn et al., 2023; Zhang et al., 2024), we consider input vectors generated through a Markovian transition kernel with parameters $p_{01}, p_{10}$ from given initial distributions. (c) The global minimizer of a linear self-attention model parameterzied by projection and attention weight matrices $P, Q$ exhibits a distinct structure compared to the ICL for linear task (Proposition 1, 2). The yellow region indicates the nontrivial portion of the global minimum of the Tranformer model parameters for ICL in linear tasks, whereas the green region becomes nontrivial in the global minimum when applied to Markovian data.

## 3 IN-CONTEXT LEARNING OF FIRST-ORDER MARKOV CHAINS FOR LSA

In this section, we present our main results on ICL for first-order Markov chains. We theoretically characterize the loss landscape of the in-context objective function $f$, where the point-wise error $\ell$ is the square loss (i.e., $\ell(\hat{y}, y) = (\hat{y} - y)^2$). Though our objective function is the mean squared loss on the query input, framing the task as a supervised regression problem, the inputs and outputs are related through a Markov chain with temporal dependencies. We analyze length-2 and arbitrary-length in-context Markov chains. For the length-2 case, we provide explicit expressions for the global minimizers. For arbitrary-length Markov chains, we derive a tight bound for the global minimum. Additionally, we provide an interpretation of the forward pass of $\text{TF}_L$ as an optimization algorithm.

## 3.1 GLOBAL MINIMUM FOR SINGLE-LAYER TRANSFORMER

For a single-layer transformer $\text{TF}_1$, we construct $(P_0, Q_0)$ to achieve a global minimum of the population loss in equation 8. The key parameters influencing the output of $\text{TF}_1$ are the last row of $P_0$ and the first $d$ columns of $Q_0$. The remaining entries are irrelevant, as the transformer output is defined solely as the bottom-right entry of $Z_1$, and the mask matrix zeros out the last column of $Q_0$. Thus, it suffices to optimize over the following subset of $P_0$ and $Q_0$:

$$P_0 = \begin{bmatrix} 0_{d \times (d+1)} \\ b^\top \end{bmatrix} \quad Q_0 = [A \quad 0_{d+1}] \tag{9}$$

where $b \in \mathbb{R}^{d+1}, A \in \mathbb{R}^{(d+1) \times d}$. Throughout this section, we assume that $P_0$ and $Q_0$ have the above format and refer to them as $P, Q$ for simplicity. The following result derives the analytic solution of a global minimizer for $f(P, Q)$ for length-2 Markov chains.

**Proposition 1** (*Global minima for i.i.d. in-context initial states*). *Consider the in-context learning of length-2 Markov chains $\{(x_i, y_i)\}_{i=1}^n$ ($x_i, y_i \in \{0, 1\}$) with transition probabilities $p_{01}, p_{11} \sim U(0, 1)$. Suppose the initial states $x_i$ are i.i.d. sampled from $Bernoulli(p)$ for some constant $p \in (0, 1)$.*

*Let $X^* := H^{-1} \begin{bmatrix} p^2/2 & p^2/3 & p^2/12 + p/4 \end{bmatrix}^\top$, where $H$ is a symmetric matrix defined as follows (repeating entries in the lower half triangle are omitted)*

$$H := p \begin{bmatrix} p/n + (n-1)p^2/n & p/2n + (n-1)p^2/2n & p/2 \\ & p/2n + (n-1)p^2/3n & p/2n + (n-1)\left(p/4 + p^2/12\right)/n \\ & & 1/2n + (n-1)\left(1/3 - p/6 + p^2/6\right)/n \end{bmatrix}.$$

*Then the following choice of parameters*

$$P = \begin{bmatrix} 0 & 0 \\ 1 & \frac{X_2^* \pm \sqrt{X_2^{*2} - 4X_1^* X_3^*}}{2} \end{bmatrix} \quad Q = \begin{bmatrix} X_1^* & 0 \\ X_2^* - \frac{X_1^* X_2^* \pm X_1^* \sqrt{X_2^{*2} - 4X_1^* X_3^*}}{2} & 0 \end{bmatrix} \tag{10}$$

*is a global minimizer of $f(P, Q)$, where $X_i^*$ is the $i$th element of $X^*$.*

See section D.1 for the proof of Proposition 1. The Markovian data requires all key model parameters to be nontrivial, unlike in-context linear tasks with zero-mean Gaussian feature and task vectors, which result in a sparser structure where the first $d$ entries of $b$ and the last row of $A$ is zero (Ahn et al., 2023; Huang et al., 2023; Zhang et al., 2024).

The independence assumption on the initial states in Proposition 1 can be removed, and we reach the following conclusion on the global minima of $f(P, Q)$, which have the same structure as the i.i.d. case.

**Proposition 2** (*Global minima for generalized in-context initial states distribution*). *Consider the in-context learning of length-2 Markov chains $\{(x_i, y_i)\}_{i=1}^n$ ($x_i, y_i \in \{0, 1\}$) with transition probabilities $p_{01}, p_{11} \sim U(0, 1)$. Suppose the initial states $x_i$ are sampled from $Bernoulli(p)$ for some constant $p \in (0, 1)$. Let $c_1 = \sum_{i=1}^n \mathbb{E}[x_i x_{n+1}], c_2 = \sum_{i=1}^n \sum_{j=1, j \neq i}^n \mathbb{E}[x_i x_j x_{n+1}]$.*

*We define $X^*$ as $X^* := H^{-1} \begin{bmatrix} c_1/2n & c_1/3n & p/4 + c_1/12n \end{bmatrix}$, where $H$ is a symmetric matrix defined as follows (repeating entries in the lower half triangle are omitted)*

$$H := \begin{bmatrix} c_1/n^2 + c_2/n^2 & c_1/2n^2 + c_2/2n^2 & c_1/2n \\ & c_1/2n^2 + c_2/3n^2 & (n+1)c_1/4n^2 + c_2/12n^2 \\ & & (2n+1)p/6n - (n-1)c_1/6n^2 + c_2/6n^2 \end{bmatrix}.$$

*(repeating entries in the lower half triangle are omitted)*

*Then by substituting $X^*$ into equation 10 gives a global minimizer of $f(P, Q)$.*

The proof for Proposition 2 is deferred to section D.2. Moreover, by relaxing the restriction on the length of the Markov chain, we obtain the following result that bounds the global minimum. We introduce a reparameterization $\phi$ which maps from the model parameter space to $\mathbb{R}^{dm}$, where $m = \frac{(d+2)(d+1)}{2}$:

$$\phi(P, Q)_r = X_r = \begin{cases} A_{i,j} b_{j'} + A_{j',j} b_{i'} & \text{for } i' \in [d+1], j' > i' \\ A_{i',j} b_{j'} & \text{for } i' \in [d+1], j' = i' \end{cases} \tag{11}$$

Here $\phi(\cdot)_r$ is the $r$th entry of the resulting vector, with $r = (j-1)m + i'(d+1) + j'$ and $A_{i,j}$ denotes the $(i,j)$-th entry of $A$ and $b_i$ denotes the $i$th element of $b$.

We verify in section D.3 that $f$ can be expressed in terms of $X$. Let $\tilde{f} : \mathbb{R}^{dm} \to \mathbb{R}$ denote the reparameterized objective s.t. $\tilde{f}(\phi(P,Q)) = f(P,Q)$. In Lemma 3, we prove that the reparameterized objective $\tilde{f}(X)$ is strictly convex. Let $X^*$ denote the global minimizer of $\tilde{f}$. Below, we present the bounds for the global minimum values for arbitrary-length in-context Markov chains.

**Theorem 1** (*Bound for global minimum for arbitrary-length Markov chains*). *We define a mapping $\psi$ that projects $X \in \mathbb{R}^{dm}$ to the parameter space: $\psi(X) = argmin_{P,Q}\|\phi(P,Q) - X\|_2^2$. Here, $\psi$ finds a parameter set that maps to the closest point to $X$ under $\phi$. $\psi(X)$ is the preimage of $X$ under $\phi$, if such a preimage exists. Let $f^*$ be the global minimum of $f$. Then $\tilde{f}(X^*) \leq f^* \leq f(\psi(X^*))$.*

Please refer to section D.3 for the proof of Theorem 1 and an example of ICL for length-3 Markov chains, where the optimal configuration of $(P,Q)$ exhibits a similarly dense structure as in the length-2 case.

## 3.2 TRANSFORMERS IMPLEMENT MULTI-OBJECTIVE OPTIMIZATION

Our goal is to find an objective function that involves the linear prediction $w^\top x_i$ for some $w \in \mathbb{R}^d$ such that the preconditioned gradient descent over this objective is equivalent to the forward pass of a multilayer LSA. To align the dimensions, we modify the sparsity condition on the attention weight matrix $Q$ by zeroing out its last row. This allows us to derive a function $R : \mathbb{R}^d \to \mathbb{R}^{d+1}$ whose Jacobian matrix is $Z_l Z_l^\top \begin{bmatrix} -\bar{A}_l \\ 0 \end{bmatrix}$. In particular, we study the subset of LSA configurations with the following sparsity constraint:

$$P = \begin{bmatrix} 0_{1\times(d+1)} \\ b_l \end{bmatrix} \quad Q = \begin{bmatrix} -\bar{A}_l & 0_d \\ 0_{1\times(d+1)} & 0 \end{bmatrix} \tag{12}$$

The following result shows that to learn arbitrary-length Markov chains in-context, a multilayer transformer implements gradient descent, preconditioned by $b_l, \bar{A}_l$, to optimize multiple objectives simultaneously.

**Proposition 3** (*Forward pass as minimizing multiple objectives*). *Consider the $L$-layer transformer parameterzed by $b_l, A_l = \begin{bmatrix} -\bar{A}_l \\ 0_{1\times d} \end{bmatrix}$ where $b_l \in \mathbb{R}^{d+1}, \bar{A}_l \in \mathbb{R}^{d\times d}$ for $l \in [L]$. Let $y_{n+1}^{(l)}$ be the bottom-right entry of the $l$th layer output. Then $y_{n+1}^{(l)} = \langle w_l^{gd}, x_{n+1}\rangle$ where $w_l^{gd}$ is iteratively defined as follows: $w_0^{gd} = 0$ and*

$$w_{l+1}^{gd} = w_l^{gd} - b_l^\top \nabla R(\theta)\bar{A}_l \quad where \; R(w) = \frac{1}{n}\sum_{i=1}^n \begin{bmatrix} -x_i \otimes \langle w, x_i\rangle \\ (\langle w, x_i\rangle - y_{n+1})^2 \end{bmatrix}$$

Proposition 3 does not involve taking the expectation of the objective; instead, it holds for an arbitrary instance of the prompt, assuming that the global minimizer satisfies the sparsity constraint specified in equation 12, which ensures dimensional alignment necessary for the derivation. The multi-objective problem involves the square loss and $d$ linear functions. The model parameters balance the optimization among these objectives, seeking to minimize the square loss within the subspace of $w$ that maximizes $x_{i,j}w_i^\top x_i$ ($j \in [d]$). Note that in ICL for linear tasks, the forward pass is equivalent to optimizing a single objective (i.e., the square loss). However, in the Markovian case, the first $d$ entries of the optimal model parameter $b$ is nonzero, preventing the linear objectives in $R$ from being canceled out.

**Remark 1.** *When the point-wise loss $\ell(\cdot, \cdot)$ in the ICL objective equation 8 is cross-entropy loss, the objective can be written as the sum of the expected KL-divergence between the predicted probability and the transition probability $\mathbb{E}_{\{x_i,y_i\}_{i=1}^{n+1}, p_{01}, p_{10}}[D_{KL}(\mathbb{P}(y_{n+1} \mid x_{n+1})\|\mathbb{P}_{P,Q}(y_{n+1} \mid x_{n+1}, \{x_i, y_i\}_{i=1}^n))]$ and entropy rate $\mathbb{E}_{x_{n+1}, p_{01}, p_{10}}[H(y_{n+1}|x_{n+1})]$, where $H(y_{n+1}|x_{n+1}) = -\sum_{s\in\mathcal{S}} \mathbb{P}(y_{n+1} = s|x_{n+1})\log\mathbb{P}(y_{n+1} = s \mid x_{n+1})$ (Makkuva et al., 2024). In this case, a global minimum equals the expected entropy rate, since $D_{KL}(\cdot\|\cdot) \geq 0$ (Thomas & Joy, 2006). We empirically demonstrate the convergence of ICL training to the entropy rate in section 4.*

## 3.3  EXPERIMENTAL VALIDATIONS

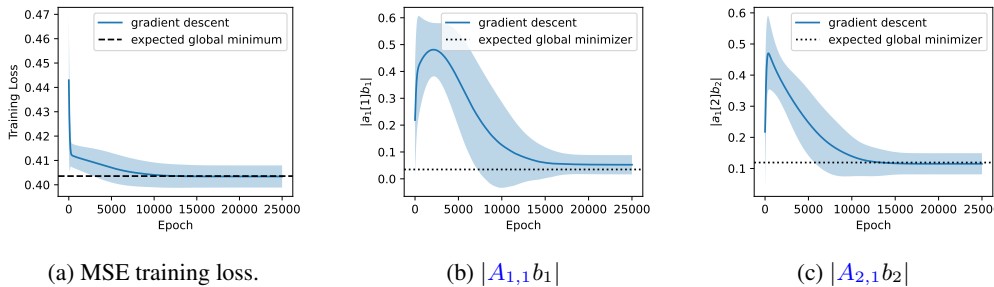

(a) MSE training loss.    (b) $|A_{1,1}b_1|$    (c) $|A_{2,1}b_2|$

Figure 2: (a) Training loss with respect to epochs for length-2 Markov chains. The dashed line represents the theoretical global minimum. (b-c) The norms of the product of two pairs of coupled parameters. Dotted lines denote minimizer of the population loss in the limit of infinite in-context examples.

In this section, we empirically validate the theoretical insights of our framework and analyze the behavior of transformers in handling Markovian dynamics. We focus on training an LSA model on length-2 binary Markov chains and examine its convergence to global minima, the impact of prompt length and initial-state distribution on global optima.

**Training and data generation.**    We optimize the following empirical objective with $B = 10K$ prompts, $n = 100$ in-context samples, and initial states sampled from $Bernoulli(0.3)$:

$$\hat{f}(P, Q) = \frac{1}{B} \sum_{k=1}^{B} (\hat{y}_{n+1}^{(k)} - y_{n+1}^{(k)})^2 \tag{13}$$

where $\hat{y}_{n+1}^{(k)}, y_{n+1}^{(k)}$ are the prediction and true labels for the query in the $k$th prompt. We apply gradient descent with a fixed step size of 0.07 for 25K epochs, initializing parameters from $U[0,1]$, and repeat this process 50 times.

**Convergence analysis.**    To form a prompt, we first sample the initial states of each in-context sequence independently from a Bernoulli distribution with parameter $p = 0.3$. Then, we sample the transition probabilities $p_{01}$ and $p_{10}$ from a uniform distribution $U(0,1)$ and generate the subsequent state for each sequence, constituting $n + 1$ length-2 Markov chains. In this case, the model parameters are $A \in \mathbb{R}^{2 \times 1}, b \in \mathbb{R}^2$. Fig. 2a shows the convergence of loss to a critical point, which aligns with the theoretical global minimum. From Fig. 2b, 2c, we observe that $A_{1,1}b_1$ and $A_{2,1}b_2$ converge to nontrivial values, indicating that $b_1$ and $A_{2,1}$ (corresponding to the green region in Fig. 1d) are nonzero. On the contrary, for ICL of linear tasks, the two terms tend to vanish, as shown by Ahn et al. (2023); Zhang et al. (2024).

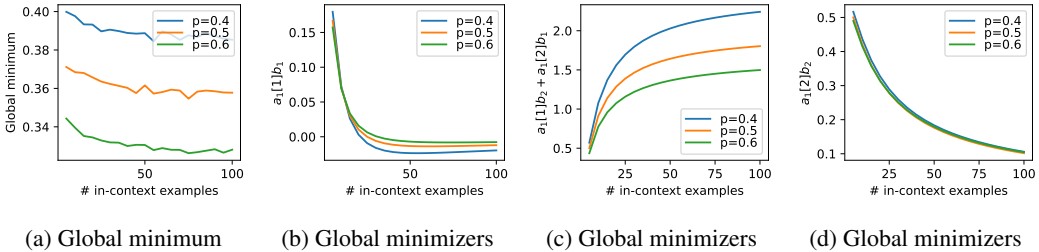

(a) Global minimum    (b) Global minimizers    (c) Global minimizers    (d) Global minimizers

Figure 3: Global minimum and optimizers versus the number of in-context samples.

**Analysis on prompt length.**    We examine the structure of the global optima when varying the in-context sample size $n$. As shown in Fig. 3a, the global minimum drops as we enlarge $n$, with

an overall smaller error for greater initial probability of sampling 1, i.e., $p$. From Fig. 3b,3c,3d, we observe the optimal $A_{1,1}b_1$ and $A_{2,1}b_2$ converge to a trivial number, approaching the optimal structure for the linear tasks with zero-mean Gaussian in-context samples.

## 4 ADDITIONAL EXPERIMENTS

Focusing on first-order binary Markov chains, we analyze the behavior of more complex transformers trained with mean squared error (MSE). Additionally, we investigate the in-context performance of transformers trained with cross-entropy loss, as detailed in the Appendix C [1].

**Data generation.** Each data sample, or a prompt, consists of $n$ sequences with length 4. To generate a prompt, we first sample the initial states of each in-context sequence indepednently from $Bernoulli(0.5)$. Then, we sample transition probabilities $p_{01}, p_{10}$ from $U(0,1)$ and iteratively generate the subsequent states for each sequence, assuming they are governed by the same Markov kernel, i.e., $\{x_i\}_{i=1}^n \sim (\pi_1 = [0.5, 0.5], P(p_{01}, p_{10}))$. Both training and testing prompts are sampled from the same distribution.

**Model and training.** We adopt architectures based on GPT-2-blocks. We consider three configurations of (embedding dimension, number of transformer blocks, number of heads), inspired by Li et al. (2024c): (i) tiny: $(64, 3, 2)$, (ii) small: $(128, 6, 4)$, (iii) standard: $(256, 12, 8)$. The models are optimized by Adam over 50K epochs with learning rate 0.0001. For each epoch, we randomly generate 64 data samples to train the model parameters. To ensure high prediction performance given any length-$n'$ prompt ($n' \in [n]$), we train on the average of the error over different prompt lengths from 1 through $n$ and update $n$ from 26 to 101 during training.

**Evaluation metric.** We report the accuracy of prediction. When the model is trained using MSE, we assign an integer within $\{0, 1\}$ that is closest to the transformer output to be the predicted state. For binary states, if the prediction is greater than 0.5, we set the predicted state to be 1 and set to 0 otherwise. When trained using cross-entropy, we assign the index of the maximal normalized logit returned by the transformer to be the predicted state.

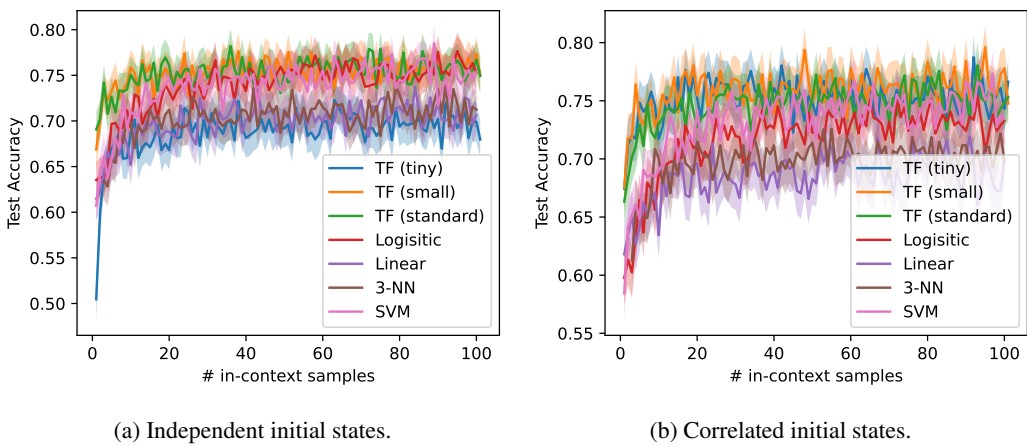

(a) Independent initial states.      (b) Correlated initial states.

Figure 4: Testing accuracy for three model configurations, compared to baseline learning algorithms.

**Transformers trained using MSE loss in-context learn next-token prediction for binary Markov chains.** We investigate the performance of trained transformer compared to baseline learning algorithms, including logistic regression, linear regression, 3-Nearest Neighbors (3-NN), and Support Vector Machine (SVM), when the number of in-context samples vary from 1 to 100. Fig. 4a,4b demonstrate the test accuracy for independent and correlated initial states. The accuracy is averaged over 1280 prompts, where the shaded region denotes 90% confidence intervals computed using

---

[1]Our code is available at `https://anonymous.4open.science/r/Markov-ICL-8351`

1000 bootstraps. The result implies that the trained transformers with small or standard size have comparable performance with SVM and logistic regression and better than the simple baseline 3-NN, while the test performance for tiny is slightly worse than its larger counterparts. While model size has a positive impact on the performance, once it reaches a threshold, the improvement is marginal. The similarity between the performance of TF and linear regression is consistent with Proposition 3, which states that the forward of trained TF optimizes a multi-objective problem including linear regression.

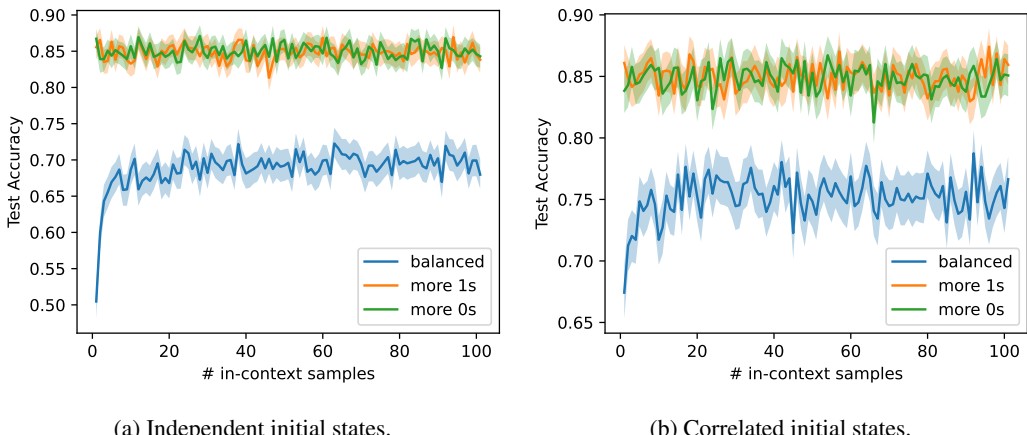

(a) Independent initial states.        (b) Correlated initial states.

Figure 5: Test accuracy with respect to the number of in-context samples, with balanced, more or less 1s.

**Entropy rate affects performance.** We explore how biased transition probabilities affect performance. In Fig. 5, we train the tiny transformer on Markov chains containing either balanced, more, or less 1s. This is controlled by drawing the transition probabilities $p_{\cdot 1}$ from $U(0,1)$, $U(0.7,1)$, and $U(0,0.3)$, respectively. Denote the query sequence of the $k$th prompt as $s^{(k)} \in \mathcal{S}^d$. We approximate the expected entropy rate of $s^{(k)}$ as follows:

$$\frac{1}{B} \sum_{k=1}^{B} \mathbb{P}(s_\tau^{(k)} = 1 \mid s_{\tau-1}^{(k)}) \log \frac{1}{\mathbb{P}(s_\tau^{(k)} = 1 \mid s_{\tau-1}^{(k)})} + \mathbb{P}(s_\tau^{(k)} = 0 \mid s_{\tau-1}^{(k)}) \log \frac{1}{\mathbb{P}(s_\tau^{(k)} = 0 \mid s_{\tau-1}^{(k)})}.$$

The empirical entropy rate for balanced, more and less 1s are 0.49, 0.39, and 0.39, respectively. The results show that for both i.i.d. (Fig. 5a) and correlated initial states (Fig. 5b), the performance is better when Markov chains are 'biased', since there is less entropy rate and therefore less uncertainty.

## 5 CONCLUSION

In this work, we investigate the in-context learning of next-token prediction tasks for dynamics-based sequential data. Specifically, we analyze the loss landscape of LSA models trained on in-context prompts consisting first-order binary Markov chains. Our findings demonstrate that the optimal transformers do not exhibit the sparsity condition typically observed ICL for linear tasks, indicating a unique adaptation of transformers to Markovian data. As the number of in-context examples increases, we observe that the global minima for length-2 Markov chains gradually approximate the sparse structure in the linear case. By introducing a special parameter construction with a sparsity level between the linear and Markovian scenarios, we show that multilayer transformers implement preconditioned gradient descent for a multi-objective optimization problem. This optimization aims to minimize the mean squared loss while maximizing linear functions of the observed in-context sequence. Furthermore, we empirically demonstrate that nonlinear transformers can successfully predict the next token when trained using cross-entropy loss, with the training loss converging to the expected entropy rate in this context. Potential extensions of our theoretical results include higher-order memory Markov chains, larger state spaces, and multilayer transformers with nonlinear attention mechanisms trained with cross-entropy loss.

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

## Contents

## A   COMPARATIVE ANALYSIS OF SETUPS

In this section, we further highlight the differences and significance of our proposed self-attention mechanism compared to existing works, focusing on both the Transformer model structure and the types of learning tasks and dynamics. This analysis sheds new light on the role of self-attention mechanisms in predicting the correct labels for in-context, sequence-level samples.

### A.1   SELF-ATTENTION MODELS

We train three variations of single-layer self-attention models with either linear (Ahn et al., 2023; Zhang et al., 2024) or nonlinear attention mechanism Vaswani et al. (2017) to in-context learn length-2 Markov chains using gradient descent over 10K random prompts. We omit the layer index when referencing the parameters since the model consists of only a single layer. The three versions of self-attention are defined as follows:

1. Variant 1 ($\mathrm{LSA}_{P,Q}^{(\mathrm{sparse})}$): LSA (equation 6) parameterized by sparse $P, Q$ (equation 9)

$$\begin{cases} Z_1 = Z_0 + \frac{1}{n} PZM(Z^\top QZ) \\ P, Q \in \{(\begin{bmatrix} 0 & 0 \\ b_1 & b_2 \end{bmatrix}, \begin{bmatrix} a_1 & 0 \\ a_2 & 0 \end{bmatrix} \mid a_i, b_i \in \mathbb{R}\} \end{cases}$$

2. Variant 2 ($\mathrm{LSA}_{P,Q}$): LSA (equation 6) parameterized by $P, Q$

$$\begin{cases} Z_1 = Z_0 + \frac{1}{n} PZM(Z^\top QZ) \\ P, Q \in \mathbb{R}^{2 \times 2} \end{cases}$$

3. Variant 3 ($\mathrm{NSA}_{W_{k,q,v}}$): Standard nonlinear self-attention (equation 5) parameterized by $W_k, W_q, W_v$.

$$\begin{cases} Z_1 = Z_0 + W_v ZM \cdot \mathrm{softmax}\left(Z^\top W_k^\top W_q Z\right), \ M := \begin{bmatrix} I_{n \times n} & 0 \\ 0 & 0 \end{bmatrix} \\ W_k, W_q, W_v \in \mathbb{R}^{2 \times 2} \end{cases}$$

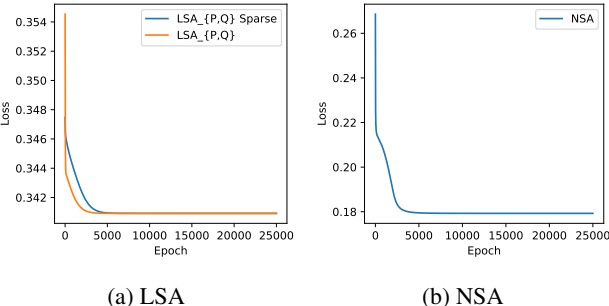

(a) LSA             (b) NSA

Figure 6: Training loss w.r.t epochs for three variants of the self-attention models, evaluated on 100 random prompts, each containing 30 in-context samples and a query sequence.

### A.1.1 LOSS CURVES

To justify the choice of the sparse parameter space, we plot the training loss curve of the above three variants in Figure 6. The loss value is the square loss for the query sequence averaged over $B$ random prompts:

$$\frac{1}{B} \sum_{\tau=1}^{B} (\hat{y}^{(\tau)} - y^{(\tau)})^2.$$

We set $B = 100$ and use 30 in-context examples for each prompt. The in-context sequences are Markov chains with initial probability 0.3 and transition probabilities $p_{01}, p_{10}$ sampled from $U(0,1)$. The results demonstrate that the loss curves under variant 1 and 2 converge to nearly the same value, indicating that the sparse and dense parameter matrices perform equivalently for LSA.

### A.1.2 ATTENTION MAPS

We visualize the attention scores and weights at convergence for three variants of the self-attention model in the plots below. We use $B = 10K$ prompts to train the first two variants to approximate their expected performance. Figure 7 displays the pairwise attention scores averaged across all random prompts. In all cases, the scores are predominantly concentrated along the diagonal, highlighting a strong emphasis on self-attention. Meanwhile, the off-diagonal entries show more evenly distributed scores, indicating a broader allocation of attention across the sequence.

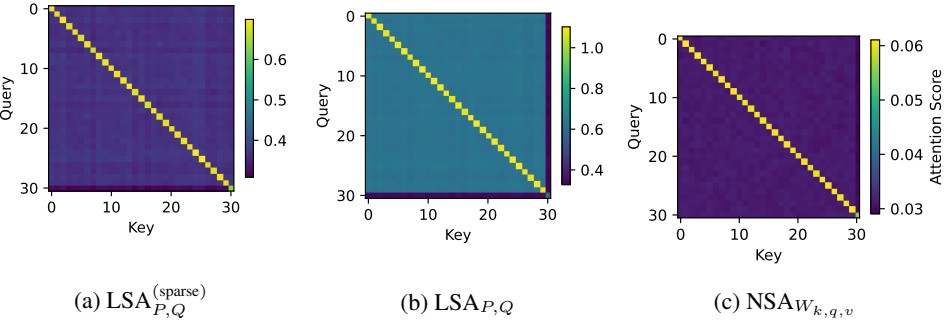

(a) $\text{LSA}_{P,Q}^{(\text{sparse})}$         (b) $\text{LSA}_{P,Q}$         (c) $\text{NSA}_{W_{k,q,v}}$

Figure 7: Attention scores at convergence, averaged over 10K prompts in (a) and (b), and 100 prompts in (c).

### A.1.3 PROJECTION AND ATTENTION WEIGHTS

In Figure 8, we show the weight matrices $P$ and $Q$ in the single-layer LSA for both sparse and nonsparse parameter space. When searching within the nonsparse parameter space, all entries are

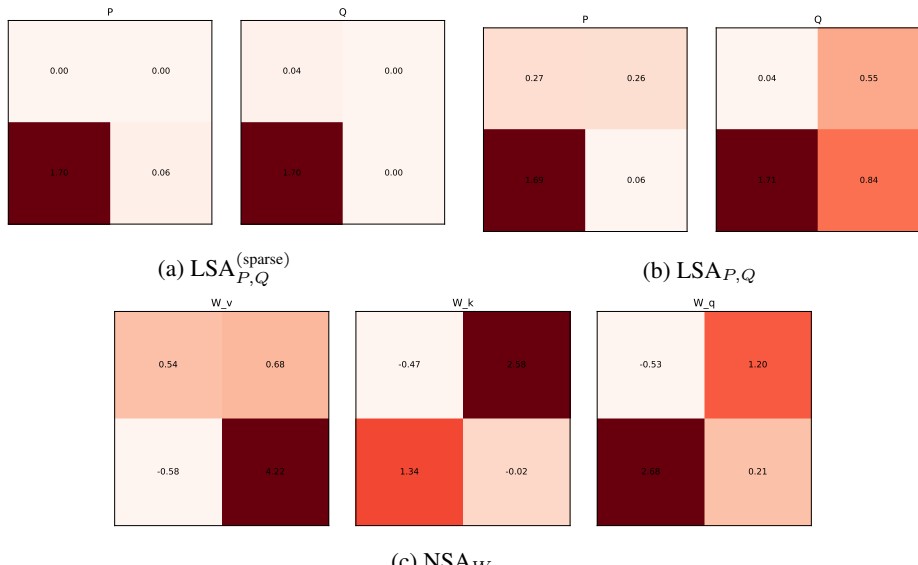

(a) $\text{LSA}_{P,Q}^{(\text{sparse})}$         (b) $\text{LSA}_{P,Q}$

(c) $\text{NSA}_{W_{k,q,v}}$

Figure 8: Projection and attention weight matrices trained using gradient descent for three variants of the self-attention model.

nontrivial at convergence. The bottom-left entry of $Q$ is dominant in both settings. This contrasts with the findings of Ahn et al. (2023), where the bottom-left entry of $Q$ converges to zero in the linear case when searched within the sparse parameter space. Our results highlight the structural differences in weight matrices under data with sequential dependence.

## A.2 ICL TASKS

We particularly compare the attention maps from three ICL tasks:

1. *ICL for Markov chain with sequence-level attention (this work).* In this setting, the Markov chains are generated from random Markov kernels with transition probabilities sampled from a given distribution. The goal is to predict the next token of a query sequence drawn from the same Markovian process as the in-context samples. Each sequence serves as an in-context example, with the attention mechanism applied across the sequences.

2. *ICL for Markov chains or other autoregressive structures with token-level attention (Sander et al., 2024; Nichani et al., 2024; Makkuva et al., 2024).* In this case, the same binary Markov chain is generated as in the previous setup. Here, each prompt consists of a single sequence, with each state in the sequence treated as an individual in-context example.

3. *ICL for linear regression (Ahn et al., 2023; Zhang et al., 2024).* The in-context input vectors and task vectors in the linear or i.i.d. case are sampled from Gaussian distributions: $x_i^{(\tau)} \sim \mathcal{N}(0, \Sigma)$ and $w^{(\tau)} \sim \mathcal{N}(0, \Lambda)$. where $\tau$ represents the prompt index and $i$ denotes the in-context index. The labels are defined as $y_i^{(\tau)} = \langle w^{(\tau)}, x_i^{(\tau)} \rangle$. Let $B$ denote the total number of prompts. The population loss is then defined as the square loss evaluated on the query for each prompt.

For each task, we train a GPT-2 model with 3 layers, each containing 2 attention heads, using AdamW optimization for 50K iterations. In the first two setups, we use both MSE and cross-entropy loss to perform in-context learning on length-6 Markov chains. For the third linear setup, we apply only MSE loss and set the in-context vector dimension to $d = 5$. During each iteration, we sample 64 random prompts, where each prompt consists of $n$ in-context sequences and one query sequence. The value of $n$ varies from 26 to 101 throughout training, following Garg et al. (2022).

The averaged attention scores for both loss functions are presented in Figure 9. Similar to the linear case (task 3), the attention map is mostly evenly distributed, with stronger intensity along the diagonal compared to other regions. Additionally, for task 1 and 2, some transformer layers exhibit columnwise sparsity. In the attention maps for task 2, the sub-diagonal entries are more

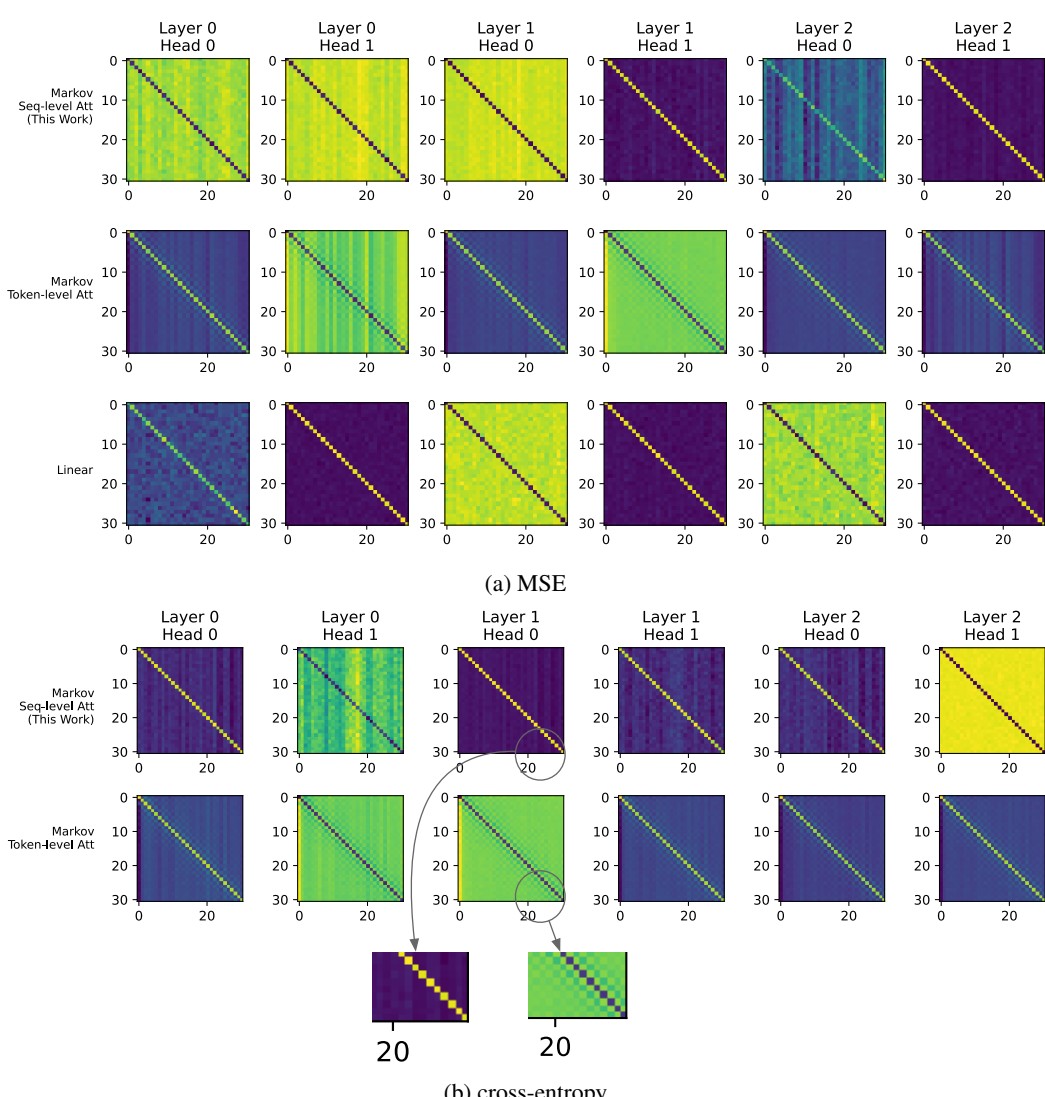

Figure 9: Attention map between in-context sequences for GPT-2 model trained using MSE and cross-entropy loss, averaged over 10K prompts. Yellow represents higher intensity and blue indicates lower intensity.

prominent compared to the other setups, reflecting the causal structure of first-order Markov chains, where each token directly influences the next. This behavior is absent in the other two setups. The sequence-level attention mechanism introduces additional challenges, as it must infer relationships between aggregated representations rather than individual tokens. This requires the model to abstract finer-grained details instead of relying on simpler patterns, such as the similarity between in-context samples and the query in the i.i.d. case, or the direct correlation between successive tokens in the second case, where attention maps primarily capture local structures. Furthermore, when the prompt construction is fixed, the attention maps trained using the two loss functions (MSE and cross entropy) display similar patterns, as both losses are designed to align predictions with the true labels.

## B    LEARNING CURVES

In this section, we numerically verify the bounds for the expected global minimum of the population loss derived in Theorem 1. We train an LSA model via gradient descent for 25K iterations on 10K prompts, each containing 100 first-order binary Markov chains and one query sequence sampled from

the same kernel. The optimization process is repeated 20 times, and the mean loss is shown as a blue curve, with the shaded region representing the standard deviation in Figure 10. The dashed black and red lines indicate the expected lower and upper bounds derived in Theorem 1, respectively, for the global minimum of the population loss in equation 8. For length-2 Markov chains, the upper and lower bounds are identical because the global minimizer $X^*$ of $\tilde{f}$ can be exactly mapped to the transformer parameter space, ensuring the existence of $P, Q$ such that $\phi(P, Q) = X^*$ (equation 11). In contrast, for length-3 Markov chains, no such $P, Q$ exists that maps to $X^*$ via $\phi$, resulting in a looser bound compared to the length-2 case, with a difference of 0.12. These numerical results also illustrate that the derived lower bound is quite tight in measuring the expected global minimum of the trained Transformer.

Note that the global minimum of $\tilde{f}$ (denoted as $\tilde{f}^*$) is always less than or equal to that of $f$. If the global minimum of $f$ were smaller than that of $\tilde{f}$, this would imply that for the global minimizers $P^*, Q^*$ of $f$, $\tilde{f}(\phi(P^*, Q^*)) = f(P^*, Q^*) < \tilde{f}^*$, which leads to a contradiction.

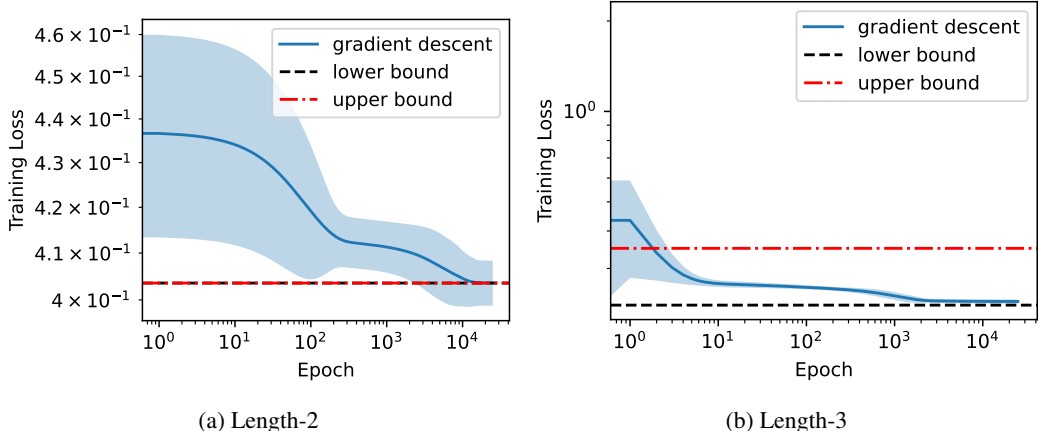

(a) Length-2                          (b) Length-3

Figure 10: Log-log plot of learning curve for LSA and the theoretical lower and upper bound for global minimum for Markov chains with length 2 and 3.

## C    Additional Experiments

In this section, we investigate the in-context performance of transformers trained with cross-entropy loss. We generate data and configure the transformer model using the same setup as in Section 4. We assess transformers trained using cross-entropy loss on predicting the next state of the query chain based on in-context sequences, with training loss and test accuracy shown in Figures 11a,11b. The loss converges to the cross-entropy rate as training progresses, aligning with Remark 1. The test accuracy of TF increases as the number of in-context examples raises, and the overall accuracy is higher than standard learning algorithms and the TF trained by MSE loss.

## D    Loss Landscape Analysis

### D.1    Proof for Independent In-context Initial States

In this section, we derive the characterization of global minima for the single layer case with binary input (Proposition 1). We begin by rewriting the loss by keeping parameters that affect the output prediction for the query $x_{n+1}$.

The input prompt is formatted as a $(d+1) \times (n+1)$ matrix:

$$Z_0 = \begin{bmatrix} x_1 & \cdots & x_n & x_{n+1} \\ y_1 & \cdots & y_n & 0 \end{bmatrix}.$$

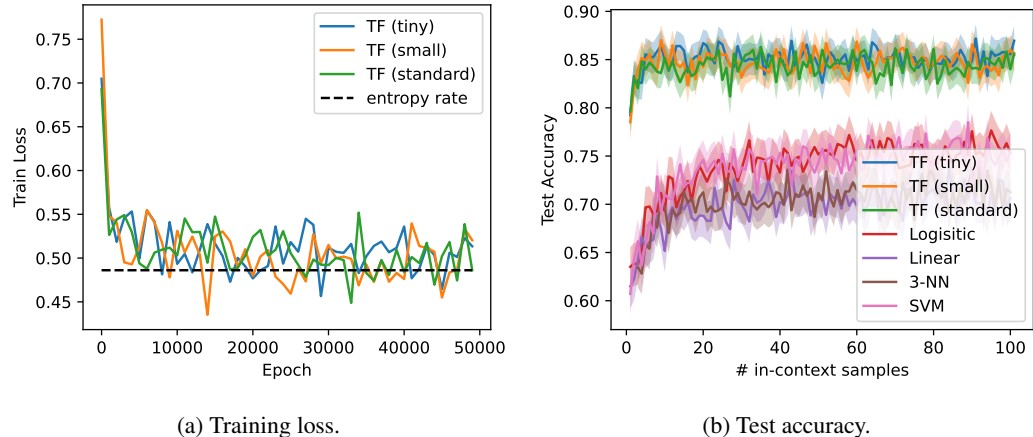

(a) Training loss.

(b) Test accuracy.

Figure 11: Training and testing performance of three transformers trained using cross-entropy loss, compared with baseline learning algorithms.

We assume $x_i \overset{i.i.d.}{\sim} Bernoulli(p)$ and let $p_{ij}$ denote the transition probability from state $i$ to $j$ ($i, j \in \mathcal{X} = \{0, 1\}$). We define the label $y_i$ to be the next state. By definition of Markov chain, the expected value of $y_i$ given $x_i$ is

$$\mathbb{E}[y_i \mid x_i, p_{01}, p_{11}] = (1 - x_i)p_{01} + x_i p_{11} = p_{01} + (p_{11} - p_{01})x_i \tag{14}$$

**Rewriting the objective function.** The in-context objective function for the single layer case is defined as:

$$f(P, Q) = \mathbb{E}_{\{x_i\}_{i=1}^{n+1}, p_{01}, p_{11}} \left[ \left( \left( Z_0 + \frac{1}{n}\text{Attn}_{P,Q}(Z_0) \right)_{d+1, n+1} - y_{n+1} \right)^2 \right] \tag{15}$$

By definition of attention (equation 6) (here $M = \begin{bmatrix} I_n & 0 \\ 0 & 0 \end{bmatrix} \in \mathbb{R}^{(n+1) \times (n+1)}$ is the mask matrix),

$$Z_0 + \frac{1}{n}\text{Attn}_{P,Q}(Z_0) = Z_0 + \frac{1}{n}PZ_0M(Z_0^\top QZ_0) = Z_0 + \frac{1}{n}P(Z_0 M Z_0^\top)QZ_0$$

$$= Z_0 + \frac{1}{n}P \left( \begin{bmatrix} x_1 & \cdots & x^{(n)} & x_{n+1} \\ y_1 & \cdots & y_n & 0 \end{bmatrix} \begin{bmatrix} I_n & 0 \\ 0 & 0 \end{bmatrix} \begin{bmatrix} x_1 & y_1 \\ \vdots & \vdots \\ x_n & y_n \\ x_{n+1} & 0 \end{bmatrix} \right) QZ_0$$

$$= Z_0 + \frac{1}{n}P \left( \begin{bmatrix} x_1 & \cdots & x_n & 0 \\ y_1 & \cdots & y_n & 0 \end{bmatrix} \begin{bmatrix} x_1 & y_1 \\ \vdots & \vdots \\ x_n & y_n \\ x_{n+1} & 0 \end{bmatrix} \right) QZ_0$$

$$= Z_0 + P \underbrace{\left( \frac{1}{n} \sum_{i=1}^{n} \begin{bmatrix} x_i^2 & x_i y_i \\ x_i y_i & y_i^2 \end{bmatrix} \right)}_{=:\mathsf{G}} QZ_0$$

The last column of the above matrix can be written as

$$\begin{bmatrix} x_{n+1} \\ 0 \end{bmatrix} + \frac{1}{n}P\mathsf{G}Q \begin{bmatrix} x_{n+1} \\ 0 \end{bmatrix}.$$

For the binary input case, $d = 1$ and $P, Q \in \mathbb{R}^{2 \times 2}$. Let $b = [b_1; b_2]^\top$ ($b \in \mathbb{R}^2$) be the last row of $P$ and $a = [a_1; a_2] \in \mathbb{R}^2$ be the first column of $Q$. The bottom-right entry of $Z_0 + \frac{1}{n}\text{Attn}_{P,Q}(Z_0)$ can

be expressed as $b^\top \mathsf{G} a x_{n+1}$. Since $f(P, Q)$ only depends on parameters $b, a$, we rewrite the objective function as

$$f(P, Q) = \mathbb{E}_{\{x_i\}_{i=1}^{n+1}, p_{01}, p_{11}} \left[ \left(b^\top \mathsf{G} a x_{n+1} - y_{n+1}\right)^2 \right] \tag{16}$$

**Reparameterization.** We further expand the term $b^\top \mathsf{G} a$ as

$$\begin{bmatrix} b_1 & b_2 \end{bmatrix} \left( \frac{1}{n} \sum_i \begin{bmatrix} x_i^2 & x_i y_i \\ x_i y_i & y_i^2 \end{bmatrix} \right) \begin{bmatrix} a_1 \\ a_2 \end{bmatrix}$$

$$= a_1 b_1 \frac{1}{n} \sum_{i=1}^n x_i^2 + (a_1 b_2 + a_2 b_1) \frac{1}{n} \sum_{i=1}^n x_i y_i + a_2 b_2 \frac{1}{n} \sum_{i=1}^n y_i^2.$$

Let $\mathsf{G}_{xx}, \mathsf{G}_{xy}, \mathsf{G}_{yy}$ denote the top-left, top-right, and bottom-right entry, respectively. For any vector $X = [X_1; X_2; X_3]$ in $\mathbb{R}^3$, we consider the following loss function

$$\tilde{f}(X) = \mathbb{E}_{\{x_i\}_{i=1}^{n+1}, p_{01}, p_{11}} \left[ \left((X_1 \mathsf{G}_{xx} + X_2 \mathsf{G}_{xy} + X_3 \mathsf{G}_{yy}) x_{n+1} - y_{n+1}\right)^2 \right] \tag{17}$$

We first derive the unique global minimum of the reparameterized loss function (equation 17) and then find the set of global minima for the original loss function (equation 15) over the space of $P, Q$.

**Lemma 1.** *Consider the in-context learning of length-2 Markov chains $\{(x_i, y_i)\}_{i=1}^n$ $(x_i, y_i \in \{0, 1\})$ with transition probabilities $p_{01}, p_{11} \sim U(0, 1)$. Suppose the initial states $x_i$ are i.i.d. sampled from $Bernoulli(p)$ for some constant $p \in (0, 1)$. Consider the reparameterized objective*

$$\tilde{f}(X) = \mathbb{E}_{\{x_i, y_i\}_{i=1}^{n+1}, p_{01}, p_{11}} \left[ \left((X_1 \mathsf{G}_{xx} + X_2 \mathsf{G}_{xy} + X_3 \mathsf{G}_{yy}) x_{n+1} - y_{n+1}\right)^2 \right] \tag{18}$$

*where $X = [X_1, X_2, X_3] \in \mathbb{R}^3$ and $y_i = (1 - x_{n+1})p_{01} + x_{n+1}p_{11}$ denotes the conditional probability observing 1 at the next state given the current state.*

*(1) The objective function $\tilde{f}$ is strictly convex.*

*(2) The global minimum $X^*$ is given as $X^* = H^{-1} \begin{bmatrix} p^2/2 & p^2/3 & p^2/12 + p/4 \end{bmatrix}^\top$, where $H$ is a symmetric matrix defined as follows*

$$H := p \begin{bmatrix} p/n + (n-1)p^2/n & p/2n + (n-1)p^2/2n & p/2 \\ & p/2n + (n-1)p^2/3n & p/2n + (n-1)\left(p/4 + p^2/12\right)/n \\ & & 1/2n + (n-1)\left(1/3 - p/6 + p^2/6\right)/n \end{bmatrix}.$$

*(omitting repeating entries in the lower half triangle).*

*Proof.* We defer the proof of *(1)* to Lemma 3. Since $\tilde{f}(X)$ is strictly convex, it has a unique global minimum that sets the gradient $\nabla \tilde{f}(X)$ to zero. To show *(2)*, we first set up the equation to evaluate the minimizer.

**Setting up equations to solve for minimizer.** The gradient of $\tilde{f}$ w.r.t. $X$ can be expressed as:

$$\nabla \tilde{f}(X) = 2 \begin{bmatrix} \mathbb{E}\left[x_{n+1}^2 \left(\mathsf{G}_{xx}^2 X_1 + \mathsf{G}_{xy}\mathsf{G}_{xx}X_2 + \mathsf{G}_{yy}\mathsf{G}_{xx}X_3\right) - x_{n+1}y_{n+1}\mathsf{G}_{xx}\right] \\ \mathbb{E}\left[x_{n+1}^2 \left(\mathsf{G}_{xx}\mathsf{G}_{xy}X_1 + \mathsf{G}_{xy}^2 X_2 + \mathsf{G}_{yy}\mathsf{G}_{xy}X_3\right) - x_{n+1}y_{n+1}\mathsf{G}_{xy}\right] \\ \mathbb{E}\left[x_{n+1}^2 \left(\mathsf{G}_{xx}\mathsf{G}_{yy}X_1 + \mathsf{G}_{xy}\mathsf{G}_{yy}X_2 + \mathsf{G}_{yy}^2 X_3\right) - x_{n+1}y_{n+1}\mathsf{G}_{yy}\right] \end{bmatrix}. \tag{19}$$

The global minimizer $X^*$ is the solution the following system:

$$\begin{bmatrix} \mathbb{E}\left[x_{n+1}^2 \mathsf{G}_{xx}^2\right] & \mathbb{E}\left[x_{n+1}^2 \mathsf{G}_{xx}\mathsf{G}_{xy}\right] & \mathbb{E}\left[x_{n+1}^2 \mathsf{G}_{xx}\mathsf{G}_{yy}\right] \\ \mathbb{E}\left[x_{n+1}^2 \mathsf{G}_{xx}\mathsf{G}_{xy}\right] & \mathbb{E}\left[x_{n+1}^2 \mathsf{G}_{xy}^2\right] & \mathbb{E}\left[x_{n+1}^2 \mathsf{G}_{xy}\mathsf{G}_{yy}\right] \\ \mathbb{E}\left[x_{n+1}^2 \mathsf{G}_{xx}\mathsf{G}_{yy}\right] & \mathbb{E}\left[x_{n+1}^2 \mathsf{G}_{xy}\mathsf{G}_{yy}\right] & \mathbb{E}\left[x_{n+1}^2 \mathsf{G}_{yy}^2\right] \end{bmatrix} \begin{bmatrix} X_1^* \\ X_2^* \\ X_3^* \end{bmatrix} = \begin{bmatrix} \mathbb{E}\left[x_{n+1}y_{n+1}\mathsf{G}_{xx}\right] \\ \mathbb{E}\left[x_{n+1}y_{n+1}\mathsf{G}_{xy}\right] \\ \mathbb{E}\left[x_{n+1}y_{n+1}\mathsf{G}_{yy}\right] \end{bmatrix}. \tag{20}$$

Next, we compute the expected values in the linear system.

**Computing RHS of equation 20.** We evaluate the three elements in RHS separately below.

1. For the first element, we have

$$\mathbb{E}_{\{x_i,y_i\}_{i=1}^{n+1},p_{01},p_{11}}\left[x_{n+1}y_{n+1}\mathsf{G}_{xx}\right]$$

$$=\mathbb{E}_{\{x_i,y_i\}_{i=1}^{n+1},p_{01},p_{11}}\left[x_{n+1}y_{n+1}\frac{1}{n}\left(\sum_{i=1}^{n}x_i^2\right)\right]$$

$$=\frac{1}{n}\sum_{i=1}^{n}\underbrace{\mathbb{E}_{\{x_i,y_i\}_{i=1}^{n+1},p_{01},p_{11}}\left[x_{n+1}y_{n+1}x_i^2\right]}_{\text{independent of } i}$$

$$=\mathbb{E}_{x_i,x_{n+1},y_{n+1}p_{01},p_{11}}\left[x_{n+1}y_{n+1}x_i^2\right]$$

$$=\mathbb{E}_{x_i,x_{n+1},p_{01},p_{11}}\left[\mathbb{E}_{y_{n+1}}\left[x_{n+1}y_{n+1}x_i^2\mid x_i,x_{n+1},p_{01},p_{11}\right]\right]$$

$$=\mathbb{E}_{x_i,x_{n+1},p_{01},p_{11}}\left[x_i^2\cdot x_{n+1}\mathbb{E}_{y_{n+1}}\left[y_{n+1}\mid x_i,x_{n+1},p_{01},p_{11}\right]\right]$$

$$\overset{(i)}{=}\mathbb{E}_{x_i,x_{n+1},p_{01},p_{11}}\left[x_i^2\cdot x_{n+1}\mathbb{E}_{y_{n+1}}\left[y_{n+1}\mid x_{n+1},p_{01},p_{11}\right]\right]$$

$$=\mathbb{E}_{x_i,x_{n+1},p_{01},p_{11}}\left[x_i^2\cdot(p_{01}x_{n+1}+(p_{11}-p_{01})x_{n+1}^2)\right]$$

$$\overset{(ii)}{=}\mathbb{E}_{x_i,x_{n+1},p_{01},p_{11}}\left[x_i^2\cdot p_{11}x_{n+1}\right]$$

$$\overset{(iii)}{=}\mathbb{E}_{p_{11}}\left[p_{11}\right]\cdot\mathbb{E}_{x_i}\left[x_i^2\right]\cdot\mathbb{E}_{x_{n+1}}\left[x_{n+1}\right]$$

$$\overset{(iv)}{=}\frac{1}{2}p^2.$$

2. Similarly, for the second element, we have

$$\mathbb{E}_{\{x_i,y_i\}_{i=1}^{n+1},p_{01},p_{11}}\left[x_{n+1}y_{n+1}\mathsf{G}_{xy}\right]$$

$$=\mathbb{E}_{x_i,y_i,x_{n+1},y_{n+1},p_{01},p_{11}}\left[x_iy_ix_{n+1}y_{n+1}\right]$$

$$\overset{(i)}{=}\mathbb{E}_{x_i,x_{n+1},p_{01},p_{11}}\left[x_i\mathbb{E}_{y_i}\left[y_i\mid x_i,p_{01},p_{11}\right]\cdot x_{n+1}\mathbb{E}_{y_{n+1}}\left[y_{n+1}\mid x_{n+1},p_{01},p_{11}\right]\right]$$

$$=\mathbb{E}_{x_i,x_{n+1},p_{01},p_{11}}\left[(p_{01}x_i+(p_{11}-p_{01})x_i^2)\cdot(p_{01}x_{n+1}+(p_{11}-p_{01})x_{n+1}^2)\right]$$

$$\overset{(ii)}{=}\mathbb{E}_{x_i,x_{n+1},p_{01},p_{11}}\left[p_{11}x_i\cdot p_{11}x_{n+1}\right]$$

$$\overset{(iii)}{=}\mathbb{E}_{p_{11}}\left[p_{11}^2\right]\cdot\mathbb{E}_{x_i}\left[x_i\right]\cdot\mathbb{E}_{x_{n+1}}\left[x_{n+1}\right]$$

$$\overset{(iv)}{=}\frac{1}{3}p^2.$$

3. The third element can be expanded as follows.

$$\mathbb{E}_{\{x_i,y_i\}_{i=1}^{n+1},p_{01},p_{11}}\left[x_{n+1}y_{n+1}\mathsf{G}_{yy}\right]$$

$$=\mathbb{E}_{x_i,y_i,x_{n+1},y_{n+1},p_{01},p_{11}}\left[x_{n+1}y_{n+1}y_i^2\right]$$

$$\overset{(i)}{=}\mathbb{E}_{x_i,x_{n+1},p_{01},p_{11}}\left[\mathbb{E}_{y_i}\left[y_i^2\mid x_i,p_{01},p_{11}\right]\cdot x_{n+1}\mathbb{E}_{y_{n+1}}\left[y_{n+1}\mid x_{n+1},p_{01},p_{11}\right]\right]$$

$$\overset{(ii)}{=}\mathbb{E}_{x_i,x_{n+1},p_{01},p_{11}}\left[\mathbb{E}_{y_i}\left[y_i\mid x_i,p_{01},p_{11}\right]\cdot(p_{11}x_{n+1})\right]$$

$$=\mathbb{E}_{x_i,x_{n+1},p_{01},p_{11}}\left[(p_{01}+(p_{11}-p_{01})x_i)\cdot(p_{11}x_{n+1})\right]$$

$$=\mathbb{E}_{x_i,x_{n+1},p_{01},p_{11}}\left[p_{01}p_{11}x_{n+1}+p_{11}^2x_ix_{n+1}-p_{01}p_{11}x_ix_{n+1}\right]$$

$$\overset{(iii)}{=}\mathbb{E}_{p_{01}}\left[p_{01}\right]\mathbb{E}_{p_{11}}\left[p_{11}\right]\mathbb{E}_{x_{n+1}}\left[x_{n+1}\right]+$$
$$\left(\mathbb{E}_{p_{11}}\left[p_{11}^2\right]-\mathbb{E}_{p_{01}}\left[p_{01}\right]\mathbb{E}_{p_{11}}\left[p_{11}\right]\right)\mathbb{E}_{x_i}\left[x_i\right]\mathbb{E}_{x_{n+1}}\left[x_{n+1}\right]$$

$$\overset{(iv)}{=}\frac{1}{4}p+\frac{1}{12}p^2.$$

**Computing LHS of equation 20.** We evaluate the expectation of the covariance of in-context examples: $\mathbb{E}[\mathsf{G}^2]$.

1.

$$\mathbb{E}_{\{x_i,y_i\}_{i=1}^{n+1},p_{01},p_{11}}\left[\mathsf{G}_{xx}^2\right]=\mathbb{E}_{\{x_i,y_i\}_{i=1}^{n+1},p_{01},p_{11}}\left[\left(\frac{1}{n}\sum_{i=1}^{n}x_i^2\right)\left(\frac{1}{n}\sum_{i=1}^{n}x_i^2\right)\right]$$

$$= \frac{1}{n^2} \left( n\mathbb{E}_{x_i}\left[x_i{}^4\right] + n(n-1)\mathbb{E}_{x_i,x_j} \underbrace{\left[x_i^2 x_j^2\right]}_{j \neq i} \right)$$

$$\stackrel{(i)}{=} \frac{1}{n^2}\left(np + n(n-1)\mathbb{E}_{x_i}\left[x_i^2\right]\mathbb{E}_{x_j}\left[x_j^2\right]\right)$$

$$\stackrel{(iv)}{=} \frac{1}{n^2}\left(np + n(n-1)p^2\right) = \frac{p}{n} + \frac{n-1}{n}p^2,$$

2.

$$\mathbb{E}_{\{x_i,y_i\}_{i=1}^{n+1},p_{01},p_{11}}\left[\mathsf{G}_{xx}\mathsf{G}_{xy}\right]$$

$$= \mathbb{E}_{\{x_i\}_{i=1}^n,\{y_i\}_{i=1}^n,p_{01},p_{11}}\left[\left(\frac{1}{n}\sum_{i=1}^n x_i^2\right)\left(\frac{1}{n}\sum_{i=1}^n x_i y_i\right)\right]$$

$$\stackrel{(ii)}{=} \mathbb{E}_{\{x_i\}_{i=1}^n,p_{01},p_{11}}\left[\left(\frac{1}{n}\sum_{i=1}^n x_i\right)\mathbb{E}_{\{y_i\}_{i=1}^n}\left[\left(\frac{1}{n}\sum_{i=1}^n x_i y_i\right)\middle|\{x_i\}_{i=1}^n,p_{01},p_{11}\right]\right]$$

$$= \mathbb{E}_{\{x_i\}_{i=1}^n,p_{01},p_{11}}\left[\left(\frac{1}{n}\sum_{i=1}^n x_i\right)\left(\frac{1}{n}\sum_{i=1}^n x_i(p_{01} + (p_{11}-p_{01})x_i)\right)\right]$$

$$= \mathbb{E}_{\{x_i\}_{i=1}^n,p_{01},p_{11}}\left[p_{01}\left(\frac{1}{n}\sum_{i=1}^n x_i\right)\left(\frac{1}{n}\sum_{i=1}^n x_i\right)\right]$$

$$+ \mathbb{E}_{\{x_i\}_{i=1}^n,p_{01},p_{11}}\left[(p_{11}-p_{01})\left(\frac{1}{n}\sum_{i=1}^n x_i\right)\left(\frac{1}{n}\sum_{i=1}^n x_i{}^2\right)\right]$$

$$\stackrel{(ii)}{=} \mathbb{E}_{\{x_i\}_{i=1}^n,p_{01},p_{11}}\left[p_{11}\left(\frac{1}{n}\sum_{i=1}^n x_i\right)\left(\frac{1}{n}\sum_{i=1}^n x_i\right)\right]$$

$$\stackrel{(iii,iv)}{=} \frac{1}{2}\mathbb{E}_{\{x_i,y_i\}_{i=1}^{n+1},p_{01},p_{11}}\left[\mathsf{G}_{xx}^2\right] = \frac{p}{2n} + \frac{n-1}{2n}p^2,$$

3.

$$\mathbb{E}_{\{x_i,y_i\}_{i=1}^{n+1},p_{01},p_{11}}\left[\mathsf{G}_{xx}\mathsf{G}_{yy}\right]$$

$$= \mathbb{E}_{\{x_i\}_{i=1}^n,p_{01},p_{11}}\left[\mathbb{E}_{\{y_i\}_{i=1}^n}\left[\left(\frac{1}{n}\sum_{i=1}^n x_i^2\right)\left(\frac{1}{n}\sum_{i=1}^n y_i^2\right)\middle|\{x_i\}_{i=1}^n,p_{01},p_{11}\right]\right]$$

$$\stackrel{(ii)}{=} \mathbb{E}_{\{x_i\}_{i=1}^n,p_{01},p_{11}}\left[\left(\frac{1}{n}\sum_{i=1}^n x_i^2\right)\mathbb{E}_{\{y_i\}_{i=1}^n}\left[\left(\frac{1}{n}\sum_{i=1}^n y_i\right)\middle|\{x_i\}_{i=1}^n,p_{01},p_{11}\right]\right]$$

$$= \mathbb{E}_{\{x_i\}_{i=1}^n,p_{01},p_{11}}\left[\left(\frac{1}{n}\sum_{i=1}^n x_i\right)\left(\frac{1}{n}\sum_{i=1}^n(p_{01} + (p_{11}-p_{01})x_i)\right)\right]$$

$$= \mathbb{E}_{\{x_i\}_{i=1}^n,p_{01},p_{11}}\left[p_{01}\left(\frac{1}{n}\sum_{i=1}^n x_i\right) + (p_{11}-p_{01})\left(\frac{1}{n}\sum_{i=1}^n x_i\right)^2\right]$$

$$\stackrel{(iii)}{=} \mathbb{E}_{p_{01}}\left[p_{01}\right]p + \mathbb{E}_{p_{01}}\left[(p_{11}-p_{01})\right]c \stackrel{(iv)}{=} \frac{p}{2},$$

4.

$$\mathbb{E}_{\{x_i,y_i\}_{i=1}^{n+1},p_{01},p_{11}}\left[\mathsf{G}_{xy}^2\right]$$

$$= \mathbb{E}_{\{x_i,y_i\}_{i=1}^n,p_{01},p_{11}}\left[\left(\frac{1}{n}\sum_{i=1}^n x_i y_i\right)\left(\frac{1}{n}\sum_{i=1}^n x_i y_i\right)\right]$$

$$= \frac{1}{n^2}\mathbb{E}_{\{x_i,y_i\}_{i=1}^{n+1},p_{01},p_{11}}\left[\sum_{i=1}^n x_i^2 y_i^2\right] + \frac{1}{n^2}\mathbb{E}_{\{x_i,y_i\}_{i=1}^{n+1},p_{01},p_{11}}\left[\sum_{i=1}^n\sum_{j=1,j\neq i}^n x_i y_i x_j y_j\right]$$

$$\stackrel{(ii)}{=} \frac{1}{n^2}\mathbb{E}_{\{x_i\}_{i=1}^n,p_{01},p_{11}}\left[\sum_{i=1}^n p_{11}x_i\right] + \frac{1}{n^2}\mathbb{E}_{\{x_i\}_{i=1}^n,p_{01},p_{11}}\left[\sum_{i=1}^n\sum_{j=1,j\neq i}^n p_{11}^2 x_i x_j\right]$$

$$=\frac{p}{2n}+\frac{(n-1)p^2}{3n},$$

5.

$$\mathbb{E}_{\{x_i,y_i\}_{i=1}^{n+1},p_{01},p_{11}}\left[\mathsf{G}_{xy}\mathsf{G}_{yy}\right]$$

$$=\mathbb{E}_{\{x_i,y_i\}_{i=1}^{n},p_{01},p_{11}}\left[\left(\frac{1}{n}\sum_{i=1}^{n}x_iy_i\right)\left(\frac{1}{n}\sum_{i=1}^{n}y_i^2\right)\right]$$

$$=\frac{1}{n^2}\mathbb{E}_{\{x_i,y_i\}_{i=1}^{n+1},p_{01},p_{11}}\left[\sum_{i=1}^{n}x_iy_i^3\right]+\frac{1}{n^2}\mathbb{E}_{\{x_i,y_i\}_{i=1}^{n+1},p_{01},p_{11}}\left[\sum_{i=1}^{n}\sum_{j=1,j\neq i}^{n}x_iy_iy_j^2\right]$$

$$\overset{(ii)}{=}\frac{1}{n^2}\mathbb{E}_{\{x_i\}_{i=1}^{n},p_{01},p_{11}}\left[\sum_{i=1}^{n}p_{11}x_i\right]+$$

$$\frac{1}{n^2}\mathbb{E}_{\{x_i\}_{i=1}^{n},p_{01},p_{11}}\left[\sum_{i=1}^{n}\sum_{j=1,j\neq i}^{n}p_{11}x_i(p_{01}+(p_{11}-p_{01})x_j)\right]$$

$$\overset{(iv)}{=}\frac{p}{2n}+\frac{n-1}{n}\left(\frac{p}{4}+\frac{p^2}{12}\right),$$

6.

$$\mathbb{E}_{\{x_i,y_i\}_{i=1}^{n+1},p_{01},p_{11}}\left[\mathsf{G}_{yy}^2\right]$$

$$=\mathbb{E}_{\{x_i,y_i\}_{i=1}^{n},p_{01},p_{11}}\left[\left(\frac{1}{n}\sum_{i=1}^{n}y_i^2\right)\left(\frac{1}{n}\sum_{i=1}^{n}y_i^2\right)\right]$$

$$=\frac{1}{n^2}\mathbb{E}_{\{x_i,y_i\}_{i=1}^{n+1},p_{01},p_{11}}\left[\sum_{i=1}^{n}y_i^4\right]+\frac{1}{n^2}\mathbb{E}_{\{x_i,y_i\}_{i=1}^{n+1},p_{01},p_{11}}\left[\sum_{i=1}^{n}\sum_{j=1,j\neq i}^{n}y_i^2y_j^2\right]$$

$$\overset{(ii)}{=}\frac{1}{n^2}\mathbb{E}_{\{x_i\}_{i=1}^{n},p_{01},p_{11}}\left[\sum_{i=1}^{n}p_{01}+(p_{11}-p_{01})x_i\right]+$$

$$\frac{1}{n^2}\mathbb{E}_{\{x_i\}_{i=1}^{n},p_{01},p_{11}}\left[\sum_{i=1}^{n}\sum_{j=1,j\neq i}^{n}(p_{01}+(p_{11}-p_{01})x_i)(p_{01}+(p_{11}-p_{01})x_j)\right]$$

$$\overset{(iv)}{=}\frac{1}{2n}+\frac{n-1}{n}\left(\frac{1}{3}-\frac{1}{6}p+\frac{1}{6}p^2\right),$$

Throughout the derivation, $(i)$ uses the fact that $\{x_j,y_j\}$ and $\{x_{j'},y_{j'}\}$ $(j'\neq j)$ are conditionally independent given $p_{01},p_{11}$; $(ii)$ holds since $x_i,y_i$ are binary random variables and $x_i^k=x_i,y_i^k=y_i$ for any integer $k$; $(iii)$ follows from the fact that $p_{01},p_{11}$ and $x_j$ $(j\in[n+1])$ are jointly independent; $(iv)$ holds because the $k$th moments of uniform distribution $U(0,1)$ and Bernoulli distribution $Bernoulli(p)$ are $\frac{1}{k+1}$ and $p$, respectively.

Since $x_{n+1}$ and $x_i$ $(i\in[n])$ are independent, we have $\mathbb{E}[x_{n+1}^2\mathsf{G}_\cdot^2]=\mathbb{E}[x_{n+1}^2]\mathbb{E}[\mathsf{G}_\cdot^2]=p\mathbb{E}[\mathsf{G}_\cdot^2]$. Hence we have the expression for $H$.

Since $\tilde{f}(X)$ is strictly convex, equation 20 has a unique solution $X^*=H^{-1}\begin{bmatrix}p^2 & p^2/3 & p^2/12+p/4\end{bmatrix}$. $\qquad\square$

**Proposition 4** (*Proposition 1 restated*). *Consider the in-context learning of length-2 Markov chains* $\{(x_i,y_i)\}_{i=1}^{n}$ $(x_i,y_i\in\{0,1\})$ *with transition probabilities* $p_{01},p_{11}\sim U(0,1)$. *Suppose the initial states* $x_i$ *are i.i.d. sampled from* $Bernoulli(p)$ *for some constant* $p\in(0,1)$.

*Let* $X^*:=H^{-1}\begin{bmatrix}p^2/2 & p^2/3 & p^2/12+p/4\end{bmatrix}^\top$, *where $H$ is a symmetric matrix defined as follows*

$$H:=p\begin{bmatrix}p/n+(n-1)p^2/n & p/2n+(n-1)p^2/2n & p/2 \\ & p/2n+(n-1)p^2/3n & p/2n+(n-1)\left(p/4+p^2/12\right)/n \\ & & 1/2n+(n-1)\left(1/3-p/6+p^2/6\right)/n\end{bmatrix}.$$

*Then the following choice of parameters*

$$P = \begin{bmatrix} 0 & 0 \\ 1 & \frac{X_2^* \pm \sqrt{X_2^{2*} - 4X_1^* X_3^*}}{2} \end{bmatrix} \quad Q = \begin{bmatrix} X_1^* & 0 \\ X_2^* - \frac{X_1^* X_2^* \pm X_1^* \sqrt{X_2^{*2} - 4X_1^* X_3^*}}{2} & 0 \end{bmatrix} \quad (21)$$

*is a global minimizer of $f(P, Q)$.*

### D.2 Proof for General Distribution of In-context Initial States

**Lemma 2.** *Consider the in-context learning of length-2 Markov chains $\{(x_i, y_i)\}_{i=1}^{n}$ ($x_i, y_i \in \{0, 1\}$) with transition probabilities $p_{01}, p_{11} \sim U0, 1$. Suppose the initial states $x_i$ are sampled from $Bernoulli(p)$ for some constant $p \in (0, 1)$. Let $c_1 = \sum_{i=1}^{n} \mathbb{E}[x_i x_{n+1}], c_2 = \sum_{i=1}^{n} \sum_{j=1, j \neq i}^{n} \mathbb{E}[x_i x_j x_{n+1}].$*

*Consider the reparameterized objective*

$$\tilde{f}(X) = \mathbb{E}_{\{x_i, y_i\}_{i=1}^{n+1}, p_{01}, p_{11}} \left[ \left( (X_1 \mathsf{G}_{xx} + X_2 \mathsf{G}_{xy} + X_3 \mathsf{G}_{yy}) x_{n+1} - y_{n+1} \right)^2 \right] \quad (22)$$

*where $X = [X_1, X_2, X_3] \in \mathbb{R}^3$ and $y_i = (1 - x_{n+1}) p_{01} + x_{n+1} p_{11}$ denotes the conditional probability observing 1 at the next state given the current state.*

*Then a global minimum is given as*

$$X^* = H^{-1} \begin{bmatrix} c_1/2n \\ c_1/3n \\ p/4 + c_1/12n \end{bmatrix} \quad (23)$$

*where*

$$H = \begin{bmatrix} c_1/n^2 + c_2/n^2 & c_1/2n^2 + c_2/2n^2 & c_1/2n \\ & c_1/2n^2 + c_2/3n^2 & (n+1)c_1/4n^2 + c_2/12n^2 \\ & & (2n+1)p/6n - (n-1)c_1/6n^2 + c_2/6n^2 \end{bmatrix}$$

*(omitting repeating entries in the lower half triangle).*

*Proof.* Since the objective function remains the same, the derivation for the equations follows from the independent in-context example case (equation 20).

**Computing RHS of equation 20 w/o assuming independence of $\{x_i\}_{i \in [n+1]}$.**

1. For the first element, we have

$$\mathbb{E}_{\{x_i, y_i\}_{i=1}^{n+1}, p_{01}, p_{11}} [x_{n+1} y_{n+1} \mathsf{G}_{xx}]$$

$$= \mathbb{E}_{\{x_i, y_i\}_{i=1}^{n+1}, p_{01}, p_{11}} \left[ x_{n+1} y_{n+1} \frac{1}{n} \left( \sum_{i=1}^{n} x_i^2 \right) \right]$$

$$= \frac{1}{n} \sum_{i=1}^{n} \mathbb{E}_{x_i, x_{n+1}, y_{n+1}, p_{01}, p_{11}} \left[ x_{n+1} y_{n+1} x_i^2 \right]$$

$$= \frac{1}{n} \sum_{i=1}^{n} \mathbb{E}_{x_i, x_{n+1}, p_{01}, p_{11}} \left[ x_{n+1} x_i \mathbb{E}_{y_{n+1}} [y_{n+1} \mid x_{n+1}, p_{01}, p_{11}] \right]$$

    # remove square, $y_i, x_j$ conditionally independent

$$= \frac{1}{n} \sum_{i=1}^{n} \mathbb{E}_{x_i, x_{n+1}, p_{11}} [p_{11} x_i x_{n+1}]$$

    # remove square

$$= \frac{1}{n} \sum_{i=1}^{n} \mathbb{E}_{p_{11}} [p_{11}] \mathbb{E}_{x_i, x_{n+1}} [x_i x_{n+1}] = \frac{1}{2n} \sum_{i=1}^{n} \mathbb{E}[x_i x_{n+1}] = \frac{1}{2n} c_1$$

    # indendence between $x_i$ and $p_{01}, p_{11}$.

2. Similarly, for the second element, we have

$$\mathbb{E}_{\{x_i,y_i\}_{i=1}^{n+1},p_{01},p_{11}}\left[x_{n+1}y_{n+1}\mathsf{G}_{xy}\right]$$

$$=\mathbb{E}_{\{x_i,y_i\}_{i=1}^{n+1},p_{01},p_{11}}\left[x_{n+1}y_{n+1}\frac{1}{n}\left(\sum_{i=1}^{n}x_iy_i\right)\right]$$

$$=\frac{1}{n}\sum_{i=1}^{n}\mathbb{E}_{x_i,y_i,x_{n+1},y_{n+1},p_{01},p_{11}}\left[x_{n+1}y_{n+1}x_iy_i\right]$$

$$=\frac{1}{n}\sum_{i=1}^{n}\mathbb{E}_{x_i,x_{n+1},p_{01},p_{11}}\left[x_{n+1}\mathbb{E}_{y_{n+1}}\left[y_{n+1}\mid x_{n+1},p_{01},p_{11}\right]\cdot x_i\mathbb{E}_{y_i}\left[y_i\mid x_i,p_{01},p_{11}\right]\right]$$

    # $y_i,y_j$ conditionally independent

$$=\frac{1}{n}\sum_{i=1}^{n}\mathbb{E}_{x_i,x_{n+1},p_{01},p_{11}}\left[(x_{n+1}p_{11})(x_ip_{11})\right]$$

    # remove square

$$=\frac{1}{3n}\sum_{i=1}^{n}\mathbb{E}[x_ix_{n+1}]=\frac{1}{3n}c_1$$

    # independence between $x_i$ and $p_{01},p_{11}$; properties of uniform distribution and joint expectation .

3. The third element can be expanded as follows.

$$\mathbb{E}_{\{x_i,y_i\}_{i=1}^{n+1},p_{01},p_{11}}\left[x_{n+1}y_{n+1}\mathsf{G}_{xy}\right]$$

$$=\mathbb{E}_{\{x_i,y_i\}_{i=1}^{n+1},p_{01},p_{11}}\left[x_{n+1}y_{n+1}\frac{1}{n}\left(\sum_{i=1}^{n}y_i^2\right)\right]$$

$$=\frac{1}{n}\sum_{i=1}^{n}\mathbb{E}_{x_i,y_i,x_{n+1},y_{n+1},p_{01},p_{11}}\left[x_{n+1}y_{n+1}y_i\right]$$

    # remove square

$$=\frac{1}{n}\sum_{i=1}^{n}\mathbb{E}_{x_i,x_{n+1},p_{01},p_{11}}\left[x_{n+1}\mathbb{E}_{y_{n+1}}\left[y_{n+1}\mid x_{n+1},p_{01},p_{11}\right]\cdot\mathbb{E}_{y_i}\left[y_i\mid x_i,p_{01},p_{11}\right]\right]$$

    # $y_i,y_j$ conditionally independent

$$=\frac{1}{n}\sum_{i=1}^{n}\mathbb{E}_{x_i,x_{n+1},p_{01},p_{11}}\left[(x_{n+1}p_{11})(p_{01}+(p_{11}-p_{01})x_i))\right]$$

    # remove square

$$=\frac{1}{n}\sum_{i=1}^{n}\mathbb{E}_{x_i,x_{n+1},p_{01},p_{11}}\left[p_{11}p_{01}x_{n+1}+(p_{11}^2-p_{11}p_{01})x_{n+1}x_i\right]$$

$$=\frac{1}{4}p+\frac{1}{12n}\sum_{i=1}^{n}\mathbb{E}[x_ix_{n+1}]=\frac{1}{4}p+\frac{1}{12n}c_1.$$

**Computing LHS of equation 20 w/o assuming independence of $\{x_i\}_{i\in[n+1]}$.** We directly present the results for the other terms, as their derivation is similar to that of the RHS in the independent case.

$$\mathbb{E}\left[x_{n+1}^2\mathsf{G}_{xx}^2\right]=\frac{1}{n^2}\sum_{i=1}^{n}\mathbb{E}[x_ix_{n+1}]+\frac{1}{n^2}\sum_{i=1}^{n}\sum_{\substack{j\neq i\\j=1}}^{n}\mathbb{E}[x_ix_jx_{n+1}]=\frac{1}{n^2}c_1+\frac{1}{n^2}c_2,$$

$$\mathbb{E}\left[x_{n+1}^2\mathsf{G}_{xx}\mathsf{G}_{xy}\right]=\frac{1}{2n^2}\sum_{i=1}^{n}\mathbb{E}[x_ix_{n+1}]+\frac{1}{2n^2}\sum_{i=1}^{n}\sum_{\substack{j\neq i\\j=1}}^{n}\mathbb{E}[x_ix_jx_{n+1}]=\frac{1}{2n^2}c_1+\frac{1}{2n^2}c_2,$$

$$\mathbb{E}\left[x_{n+1}^2\mathsf{G}_{xx}\mathsf{G}_{yy}\right]=\frac{1}{2n}\sum_{i=1}^{n}\mathbb{E}[x_ix_{n+1}]=\frac{1}{2n}c_1,$$

$$\mathbb{E}\left[x_{n+1}^2 \mathsf{G}_{xy}^2\right] = \frac{1}{2n^2}\sum_{i=1}^n \mathbb{E}[x_i x_{n+1}] + \frac{1}{3n^2}\sum_{i=1}^n \sum_{\substack{j\neq i \\ j=1}}^n \mathbb{E}[x_i x_j x_{n+1}] = \frac{1}{2n^2}c_1 + \frac{1}{3n^2}c_2,$$

$$\mathbb{E}\left[x_{n+1}^2 \mathsf{G}_{xy}\mathsf{G}_{yy}\right] = \frac{1}{2n^2}\sum_{i=1}^n \mathbb{E}[x_i x_{n+1}] + \frac{1}{n^2}\sum_{i=1}^n \sum_{\substack{j\neq i \\ j=1}}^n \frac{1}{4}\mathbb{E}[x_i x_{n+1}] + \frac{1}{12}\mathbb{E}[x_i x_j x_{n+1}] = \frac{n+1}{4n^2}c_1 + \frac{1}{12n^2}c_2.$$

$$\mathbb{E}\left[x_{n+1}^2 \mathsf{G}_{yy}^2\right] = \frac{p}{2n} + \frac{(n-1)p}{3n} + \frac{1}{n^2}\sum_{i=1}^n \sum_{\substack{j\neq i \\ j=1}}^n -\frac{1}{12}\mathbb{E}[x_i x_{n+1}] - \frac{1}{12}\mathbb{E}[x_j x_{n+1}] + \frac{1}{6}\mathbb{E}[x_i x_j x_{n+1}]$$

$$= \frac{(2n+1)p}{6n} - \frac{n-1}{6n^2}c_1 + \frac{1}{6n^2}c_2.$$

$\square$

**Proposition 5** (**Proposition 2 restated**). *Consider the in-context learning of length-2 Markov chains $\{(x_i, y_i)\}_{i=1}^n$ $(x_i, y_i \in \{0,1\})$ with transition probabilities $p_{01}, p_{11} \sim U(0,1)$. Suppose the initial states $x_i$ are sampled from $Bernoulli(p)$ for some constant $p \in (0,1)$. Let $c_1 = \sum_{i=1}^n \mathbb{E}[x_i x_{n+1}], c_2 = \sum_{i=1}^n \sum_{j=1, j\neq i}^n \mathbb{E}[x_i x_j x_{n+1}]$.*

*We define $X^*$ as $X^* := H^{-1}\left[c_1/2n \quad c_1/3n \quad p/4 + c_1/12n\right]$, where $H$ is a symmetric matrix defined as follows (repeating entries in the lower half triangle are omitted)*

$$H := \begin{bmatrix} c_1/n^2 + c_2/n^2 & c_1/2n^2 + c_2/2n^2 & c_1/2n \\ & c_1/2n^2 + c_2/3n^2 & (n+1)c_1/4n^2 + c_2/12n^2 \\ & & (2n+1)p/6n - (n-1)c_1/6n^2 + c_2/6n^2 \end{bmatrix}.$$

*(repeating entries in the lower half triangle are omitted)*

*Then by substituting $X^*$ into equation 10 gives a global minimizer of $f(P, Q)$.*

**Example 1.** *Suppose $x_{n+1} \sim Bernoulli(p)$ and $x_i \mid x_{n+1} \sim Bernoulli(g(x_{n+1}))$ for some function $g : \{0,1\} \to [0,1]$. For example, when $g(x) = (x-p)^2$, the expected values can be computed as follows.*

*For $i \in [n]$, $j = n+1$,*

$$\begin{aligned} \mathbb{E}[x_i x_{n+1}] &= \mathbb{E}_{x_{n+1}}\left[x_{n+1}\mathbb{E}_{x_i}[x_i | x_{n+1}]\right] \\ &= \mathbb{E}_{x_{n+1}}\left[x_{n+1}^3 - 2px_{n+1}^2 + p^2 x_{n+1}\right] \\ &= p - 2p^2 + p^3. \end{aligned}$$

*Therefore $c_1 = n(p - 2p^2 + p^3)$.*

$$\begin{aligned} \frac{c_2}{n(n-1)} &= \mathbb{E}[x_i x_j x_{n+1}] \\ &= \mathbb{E}_{x_{n+1}}\left[x_{n+1}\mathbb{E}_{x_i}[x_i|x_{n+1}]\mathbb{E}_{x_j}[x_j|x_{n+1}]\right] \\ &\quad \# \ x_i, x_j \text{ are conditionally ind. given } x_{n+1} \\ &= \mathbb{E}_{x_{n+1}}\left[x_{n+1}(x_{n+1}-p)^2(x_{n+1}-p)^2\right] \\ &= \mathbb{E}_{x_{n+1}}\left[x_{n+1}(x_{n+1}^2 - 2px_{n+1} + p^2)(x_{n+1}^2 - 2px_{n+1} + p^2)\right] \\ &= \mathbb{E}_{x_{n+1}}\left[x_{n+1}((1-2p)x_{n+1} + p^2)^2\right] \\ &\quad \# \text{ expected values of squares of } x_{n+1} \text{ is equivalent to that of } x_{n+1} \\ &= \mathbb{E}_{x_{n+1}}\left[(1 - 4p + 4p^2)x_{n+1}^3 + 2(1-2p)p^2 x_{n+1}^2 + p^4 x_{n+1}\right] \\ &= p - 4p^2 + 4p^3 + 2p^3 - 4p^4 + p^5 \\ &= p^5 - 4p^4 + 6p^3 - 4p^2 + p. \end{aligned}$$

## D.3 ICL FOR ARBITRARY-LENGTH MARKOV CHAINS

We recall $(x_i, y_i)$ form a binary Markov chain of length $d+1$. Assuming the initial states are sampled from $Bernoulli(p)$, the probability of $x_{i,1}$ being 1 is $p$. For $1 < j \leq d$, the probability of $x_{i,j}$

being 1, given $x_{i,j-1}$, is $p_{11}x_{i,j-1} + (1 - x_{i,j-1})p_{01}$. The probability of $y_i$ being 1, given $x_{i,d}$, is $p_{11}x_{i,d} + (1 - x_{i,d})p_{01}$.

**Reparameterization.** For general $d \geq 1$, the projection matrix $P$ and attention weight matrix $Q$ are of size $(d+1) \times (d+1)$. We write

$$P = \begin{bmatrix} 0_{d \times (d+1)} \\ b^\top \end{bmatrix} \quad Q = \begin{bmatrix} A & 0_{d+1} \end{bmatrix} \tag{24}$$

where $b^\top \in \mathbb{R}^{1 \times (d+1)}$ denote the last row of $P$ and $A \in \mathbb{R}^{(d+1) \times d}$ ($j \in [d]$) represent the first $d$ columns of $Q$. The objective function can be rewritten as:

$$f(P, Q) = \mathbb{E}_{\{x_i, y_i\}_{i=1}^{n+1}, p_{01}, p_{11}} \left[ \left( \sum_{j=1}^d b^\top \mathsf{G} a_j x_{n+1,j} - y_{n+1} \right)^2 \right], \tag{25}$$

where $x_{n+1,j}$ ($j \in [d]$) denotes the $j$th element of $x_{n+1}$. The $i$-$j$ entry of $\mathsf{G}$ ($\mathsf{G}_{i,j}$) has the following expression:

$$\mathsf{G}_{i,j} = \begin{cases} 1/n \sum_{k=1}^n x_{k,i} x_{k,j} & \text{if } i, j \in [d] \\ 1/n \sum_{k=1}^n x_{k,i} y_k & \text{if } i \in [d], j = d+1 \text{ or } i = d+1, j \in [d] \\ 1/n \sum_{k=1}^n y_k^2 & \text{if } i, j = d+1 \end{cases} \tag{26}$$

Since $\mathsf{G}$ is symmetric, to obtain an objective function with a unique global minimum, we collect model parameters that share the same coefficients $\mathsf{G}_{i,j} = \mathsf{G}_{j,i}$. We introduce a reparameterization $\phi$ which maps from the model parameter space to $\mathbb{R}^{dm}$, where $m = \frac{(d+2)(d+1)}{2}$:

$$\phi(P, Q)_r = X_r = \begin{cases} A_{i,j} b_{j'} + A_{j',j} b_{i'} & \text{for } i' \in [d+1], j' > i' \\ A_{i',j} b_{j'} & \text{for } i' \in [d+1], j' = i' \end{cases} \tag{27}$$

Here $\phi(\cdot)_r$ is the $r$th entry of the resulting vector, with $r = (j-1)m + i'(d+1) + j'$ and $A_{i,j}$ denotes the $(i, j)$-th entry of $A$ and $b_i$ denotes the $i$th element of $b$.

To simplify notation, we collapse the unique elements in $\mathsf{G}$ into a vector:

$$\mathsf{g} = \begin{bmatrix} \mathsf{G}_{1,1} & \mathsf{G}_{1,2} & \cdots & \mathsf{G}_{1,d+1} & \mathsf{G}_{2,2} & \cdots & \mathsf{G}_{d,d} & \mathsf{G}_{d,d+1} & \mathsf{G}_{d+1,d+1} \end{bmatrix}^\top. \tag{28}$$

We concatenate the parameters $X^{(j)}$ ($j \in [d]$) into a vector $X = [X^{(1)}; \ldots; X^{(d)}] \in \mathbb{R}^{dm}$ and consider the following reparameterized objective function

$$\tilde{f}(X) = \mathbb{E}_{\{x_i, y_i\}_{i=1}^{n+1}, p_{01}, p_{11}} \left[ \left( (x_{n+1} \otimes \mathsf{g})^\top X - y_{n+1} \right)^2 \right]. \tag{29}$$

**Lemma 3.** *Suppose the initial probability of the Markov chains is $\pi_1 = [1 - p, p]$ with $p \in (0, 1)$ and the transition probabilities are sampled from $U(0, 1)$. The reparameterized objective function equation 29 is strictly convex w.r.t. $X \in \mathbb{R}^{dm}$.*

*Proof.* We show the Hessian of $\tilde{f}$ w.r.t. $X$, $\mathbb{E}[x_{n+1} x_{n+1}^\top \otimes \mathsf{g}\mathsf{g}^\top]$, is positive definite. Let $w \neq 0$ be an arbitrary nontrivial vector in $\mathbb{R}^{dm}$. Let $z := x_{n+1} \otimes \mathsf{g}$. Then for any $x_{n+1} \in \{0,1\}^d$ and $\mathsf{g} \in [0,1]^m$, $w^\top \mathbb{E}[x_{n+1} x_{n+1}^\top \otimes \mathsf{g}\mathsf{g}^\top] w = w^\top \mathbb{E}[(x_{n+1} \otimes \mathsf{g})(x_{n+1} \otimes \mathsf{g})^\top] w = w^\top z z^\top w = |w^\top z|^2 \geq 0$. Since $w \neq 0$, at least one of its entry is nonzero and this entry is multiplied by one of $\{x_{n+1,j} \mathsf{G}_{i',j'} : j \in [d], i', j' \in [d+1]\}$ in the expression $w^\top z$. Take $j = \alpha, i' = \beta, j' = \gamma$. Then it suffices to find specific $\{x_i, y_i\}_{i=1}^n$ and $x_{n+1}$ s.t. $x_{n+1}[\alpha] \mathsf{G}_{\beta,\gamma} > 0$ with positive probability, i.e., $\mathbb{P}[x_{n+1,\alpha} \mathsf{G}_{\beta,\gamma}] > 0$. Since the initial probability $p \in (0, 1)$ and the transition probabilities $p_{ij}$ are nonzero, by definition of Markov chains, $\mathbb{P}[x_{n+1,\alpha} \mathsf{G}_{\beta,\gamma}]$ is the product of $p$ (or $1 - p$) and $p_{ij}$s and therefore is nonzero. Now because $w^\top(z z^\top)w \geq 0$ for all $z$ in its support and there exists at least one $z \in \mathbb{R}^{dm}$ s.t. $w^\top(z z^\top)w > 0$ and $\mathbb{P}[z] > 0$, we have $w^\top \mathbb{E}[z z^\top] w > 0$. Hence the matrix $\mathbb{E}[x_{n+1} x_{n+1}^\top \otimes \mathsf{g}\mathsf{g}]$ is positive definite and it follows that $\tilde{f}$ is strictly convex.

$\square$

**Lemma 4.** *Consider the in-context learning of length-$d+1$ ($d \geq 1$) Markov chains $\{(x_i, y_i)\}_{i=1}^{n}$ ($x_i, y_i \in \{0, 1\}$) with transition kernel $\mathsf{P} = \begin{bmatrix} p_{00} & p_{01} \\ p_{10} & p_{11} \end{bmatrix} \in (0, 1)^2$. Suppose the initial states $x_i$ are i.i.d. sampled from $Bernoulli(p)$ for some constant $p \in (0, 1)$. Consider indices $i, j \in [d]$, $i', j', k', l' \in [d+1]$ with $i' \leq j', k' \leq l'$. We denote $t_1 \leq t_2 \leq t_3 \leq t_4$ as the sorted version of $(i', j', k', l')$. Define $H \in \mathbb{R}^{dm \times dm}$ as*

$$H_{r,c} = \frac{1}{n} \mathbb{E}\left[ \left( p(\mathsf{P}^{t_1-1})_{11} + (1-p)(\mathsf{P}^{t_1-1})_{01} \right) (\mathsf{P}^{t_2-t_1})_{11} (\mathsf{P}^{t_3-t_2})_{11} (\mathsf{P}^{t_4-t_3})_{11} \right] +$$

$$\frac{n-1}{n} \mathbb{E}\left[ \left( p(\mathsf{P}^{i'-1})_{11} + (1-p)(\mathsf{P}^{i'-1})_{01} \right) (\mathsf{P}^{j'-i'})_{11} \right.$$

$$\left. \left( p(\mathsf{P}^{k'-1})_{11} + (1-p)(\mathsf{P}^{k'-1})_{01} \right) (\mathsf{P}^{l'-k'})_{11} \right]$$

*where $r = (i-1)m + j' + \sum_{\tau=0}^{i'-2} d + 1 - \tau$, $c = (j-1)m + l' + \sum_{\tau=0}^{k'-2} d + 1 - \tau$.*

*Define $b \in \mathbb{R}^{dm}$ as*

$$b_{(j-1)m+j'+\sum_{\tau=0}^{i'-2} d+1-\tau} = \mathbb{E}\left[ \left( p(\mathsf{P}^{j-1})_{11} + (1-p)(\mathsf{P}^{j-1})_{01} \right) (\mathsf{P}^{d+1-j})_{11} \right.$$

$$\left. \left( p(\mathsf{P}^{i'-1})_{11} + (1-p)(\mathsf{P}^{i'-1})_{01} \right) (\mathsf{P}^{j'-i'})_{11} \right].$$

*The global minimum $X^* \in \mathbb{R}^{dm}$ of the objective function described in equation 29 equals $X^* = H^{-1}b$.*

*Proof.* Setting the gradient of equation 29 w.r.t. $X$ to zero, we have

$$\mathbb{E}\left[ x_{n+1} x_{n+1}^\top \otimes \mathsf{g}\mathsf{g}^\top \right] X = \mathbb{E}\left[ y_{n+1} \left( x_{n+1} \otimes \mathsf{g} \right) \right]. \tag{30}$$

where $\otimes$ denotes the Kronecker product.

**Evaluating LHS of equation 30:** $\mathbb{E}\left[ x_{n+1,i} x_{n+1,j} \mathsf{G}_{i',j'} \mathsf{G}_{k',l'} \right]$ **with $i, j \in [d]$, $i \leq j$, $i', j', k, l' \in [d+1]$, and $i' \leq k', j' \leq l'$.** Let $\mathsf{P} = \begin{bmatrix} p_{00} & p_{01} \\ p_{10} & p_{11} \end{bmatrix}$ denote the transition probability matrix and $\pi = [1-p, p]$ the initial marginal probability. Further, $(\mathsf{P}^k)_{ij}$ ($i, j \in \{0, 1\}$) denotes the specific entry of $\mathsf{P}$ raised to the power of $k$. Then

$$\mathbb{E}\left[ x_{n+1,i} x_{n+1,j} \mathsf{G}_{i',j'} \mathsf{G}_{k',l'} \right] = \mathbb{E}\left[ x_{n+1,i} x_{n+1,j} \right] \mathbb{E}\left[ \mathsf{G}_{i',j'} \mathsf{G}_{k',l'} \right],$$

due to the fact that $x_i$ ($i \in [n]$) and $x_{n+1}$ are independent and $\mathsf{G}$ contains in-context samples only. We then evaluate the two terms $\mathbb{E}\left[ x_{n+1,i} x_{n+1,j} \right], \mathbb{E}\left[ \mathsf{G}_{i',j'} \mathsf{G}_{k',l'} \right]$ separately.

▶ For $i \leq j$, the probability of $x_{n+1,i} = x_{n+1,j} = 1$ is equivalent to that of $x_{n+1,i} = 1$ conditioned on $x_{n+1,1}$ multiplied by the probability of $x_{n+1,j} = 1$ conditioned on $x_{n+1,i} = 1$. Therefore,

$$\mathbb{E}\left[ x_{n+1,i} x_{n+1,j} \right] = \mathbb{E}\left[ \mathbb{P}[x_{n+1,i} = x_{n+1,j} = 1] \right.$$

$$= \mathbb{E}\left[ \left( p(\mathsf{P}^{i-1})_{11} + (1-p)(\mathsf{P}^{i-1})_{01} \right) (\mathsf{P}^{j-i})_{11} \right].$$

▶ We temporarily let $x_{k,d+1} = y_k$ for $k \in [n]$. For $i', j', k', l' \in [d+1]$ and $i' \leq j', k' \leq l'$, we have

$$\mathbb{E}\left[ \mathsf{G}_{i',j'} \mathsf{G}_{k',l'} \right]$$

$$= \mathbb{E}\left[ \left( \frac{1}{n} \sum_{k=1}^{n} x_{k,i'} x_{k,j'} \right) \left( \frac{1}{n} \sum_{k=1}^{n} x_{\kappa,k'} x_{\kappa,l'} \right) \right]$$

$$= \frac{1}{n^2} \mathbb{E}\left[ \left( \sum_{k=1}^{n} x_{k,i'} x_{k,j'} \right) \left( \sum_{\kappa=1}^{n} x_{\kappa,k'} x_{\kappa,l'} \right) \right]$$

$$= \frac{1}{n^2} \mathbb{E}\left[ \sum_{k=1}^{n} \sum_{\kappa=1}^{n} x_{k,i'} x_{k,j'} x_{\kappa,k'} x_{\kappa,l'} \right]$$

$$= \frac{1}{n^2} \mathbb{E}\left[\sum_{k=1}^n x_{k,i'} x_{k,j'} x_{\kappa,k'} x_{\kappa,l'}\right]$$

$$+ \frac{1}{n^2} \mathbb{E}\left[\sum_{k=1}^n \sum_{\kappa=1,\kappa \neq k}^n x_{k,i'} x_{k,j'} x_{\kappa,k'} x_{\kappa,l'}\right].$$

The summands in the first term, in the case of $j' \leq k'$, has the following form. The remaining orderings of $i', j', k', l'$ can be computed in a similar manner.

$$\mathbb{E}\left[\sum_{k=1}^n x_{k,i'} x_{k,j'} x_{k,k'} x_{k,l'}\right]$$

$$= \mathbb{E}\left[\sum_{k=1}^n \mathbb{P}\left[x_{k,i'} = x_{k,j'} = x_{k,k'} = x_{k,l'} = 1\right]\right]$$

$$= n \mathbb{E}\left[\left(p(\mathsf{P}^{i'-1})_{11} + (1-p)(\mathsf{P}^{i'-1})_{01}\right)(\mathsf{P}^{j'-i'})_{11}(\mathsf{P}^{k'-j'})_{11}(\mathsf{P}^{l'-k'})_{11}\right].$$

The summands in the second term can be calculated as below.

$$\mathbb{E}\left[\sum_{k=1}^n \sum_{\kappa=1,\kappa \neq k}^n x_{k,i'} x_{k,j'} x_{\kappa,k'} x_{\kappa,l'}\right]$$

$$= \mathbb{E}\left[\sum_{k=1}^n \sum_{\kappa=1,\kappa \neq k}^n \mathbb{P}\left[x_{k,i'} = x_{k,j'} = 1\right] \mathbb{P}\left[x_{\kappa,k'} = x_{\kappa,l'} = 1\right]\right]$$

$$= n(n-1)\mathbb{E}\left[\left(p(\mathsf{P}^{i'-1})_{11} + (1-p)(\mathsf{P}^{i'-1})_{01}\right)(\mathsf{P}^{j'-i'})_{11}\right.$$

$$\left.\left(p(\mathsf{P}^{k'-1})_{11} + (1-p)(\mathsf{P}^{k'-1})_{01}\right)(\mathsf{P}^{l'-k'})_{11}\right].$$

**Evaluating RHS of equation 30:** $\mathbb{E}[x_{n+1,j}y_{n+1}\mathsf{G}_{i',j'}]$ **with** $i' \leq j'$.

$$\mathbb{E}[x_{n+1,j}y_{n+1}\mathsf{G}_{i',j'}] = \frac{1}{n}\sum_{k=1}^n \mathbb{E}\left[x_{n+1,j}y_{n+1}x_{k,i'}x_{k,j'}\right]$$

$$= \frac{1}{n}\sum_{k=1}^n \mathbb{E}\left[\mathbb{P}\left[x_{n+1,j} = y_{n+1} = 1\right]\mathbb{P}\left[x_{k,i'} = x_{k,j'} = 1\right]\right]$$

$$= \mathbb{E}\left[\left(p(\mathsf{P}^{j-1})_{11} + (1-p)(\mathsf{P}^{j-1})_{01}\right)(\mathsf{P}^{d+1-j})_{11}\right.$$

$$\left.\left(p(\mathsf{P}^{i'-1})_{11} + (1-p)(\mathsf{P}^{i'-1})_{01}\right)(\mathsf{P}^{j'-i'})_{11}\right].$$

$\square$

**Theorem 2** ( *Theorem 1 restated*). *We define a mapping $\psi$ that projects $X \in \mathbb{R}^{dm}$ to the parameter space: $\psi(X) = \mathrm{argmin}_{P,Q}\|\phi(P,Q) - X\|_2^2$. Here, $\psi$ finds a parameter set that maps to the closest point to $X$ under $\phi$. $\psi(X)$ is the preimage of $X$ under $\phi$, if such a preimage exists. Let $f^*$ be the global minimum of $f$. Then $\tilde{f}(X^*) \leq f^* \leq f(\psi(X^*))$.*

*Proof.* (sketch) Let $P^*, Q^*$ denote the global minimizer corresponding to $f^*$. Since $\tilde{f}$ is strictly convex w.r.t $X \in \mathbb{R}^{dm}$ , it follows that $\tilde{f}(X^*)$ is the lower bound for any $\tilde{f}(\phi(P,Q))$, including $f^* = f(P^*,Q^*) = \tilde{f}(\phi(P^*,Q^*))$. Therefore $\tilde{f}(X^*) \leq f^*$. Similarly, since $f^*$ is smaller than any $f(P,Q)$, we have $f^* \leq f(\psi(X^*))$. $\square$

**Example 2.** *As an example, for $d = 2$, $\mathbf{g}\mathbf{g}^\top$ becomes*

$$\begin{bmatrix} \mathsf{G}_{11}^2 & \mathsf{G}_{11}\mathsf{G}_{12} & \mathsf{G}_{11}\mathsf{G}_{13} & \mathsf{G}_{11}\mathsf{G}_{22} & \mathsf{G}_{11}\mathsf{G}_{23} & \mathsf{G}_{11}\mathsf{G}_{33} \\ & \mathsf{G}_{12}^2 & \mathsf{G}_{12}\mathsf{G}_{13} & \mathsf{G}_{12}\mathsf{G}_{22} & \mathsf{G}_{12}\mathsf{G}_{23} & \mathsf{G}_{12}\mathsf{G}_{33} \\ & & \mathsf{G}_{13}^2 & \mathsf{G}_{13}\mathsf{G}_{22} & \mathsf{G}_{13}\mathsf{G}_{23} & \mathsf{G}_{13}\mathsf{G}_{33} \\ & & & \mathsf{G}_{22}^2 & \mathsf{G}_{22}\mathsf{G}_{23} & \mathsf{G}_{22}\mathsf{G}_{33} \\ & & & & \mathsf{G}_{23}^2 & \mathsf{G}_{23}\mathsf{G}_{33} \\ & & & & & \mathsf{G}_{33}^2 \end{bmatrix} \tag{31}$$

*(omitting the index-separating comma and the repeating entries in the lower half triangle).*

*After reparameterization, the objective function can be rewritten as*

$$\tilde{f}(X) = \mathbb{E}_{\{x_i, y_i\}_{i=1}^{n+1}, p_{01}, p_{11}} \left[ \left( \sum_{j=1}^2 \mathbf{g}^\top X^{(j)} x_{n+1,j} - y_{n+1} \right)^2 \right].$$

*where $X \in \mathbb{R}^{12}$ denotes the concatenation of the two vectors $X^{(1)}, X^{(2)} \in \mathbb{R}^6$. The gradient of $\tilde{f}$ w.r.t. $X$ is*

$$\nabla \tilde{f}(X) = \mathbb{E} \begin{bmatrix} (x_{n+1,1})^2 \mathbf{g}\mathbf{g}^\top & x_{n+1,1} x_{n+1,2} \mathbf{g}\mathbf{g}^\top \\ x_{n+1,2} x_{n+1,2} \mathbf{g}\mathbf{g}^\top & (x_{n+1,2})^2 \mathbf{g}\mathbf{g}^\top \end{bmatrix} X - \mathbb{E}[y_{n+1}(x_{n+1} \otimes \mathbf{g})].$$

*We obtain the global minimizer $X^*$ by solving $\nabla \tilde{f}(X^*) = 0$.*

$$X^{(1)^*} = \begin{bmatrix} -0.15 & 0.39 & 0.15 & 0.12 & 2.40 & -0.09 \end{bmatrix}$$
$$X^{(2)^*} = \begin{bmatrix} 0.07 & -0.19 & -0.07 & -0.06 & -1.20 & 0.04 \end{bmatrix}$$

*We project $X^{(1)}, X^{(2)}$ into the model parameter space.*

*Since the entires of $X^{(1)}$ are nonzero, we have $b_1 \neq b_2$*

*To verify the derivation, we plot the loss function w.r.t $X_i$, indicating the global optimizer $X_i^*$ using red dashed line in Fig. 12. The theoretical global minimizer aligns with the lowest error.*

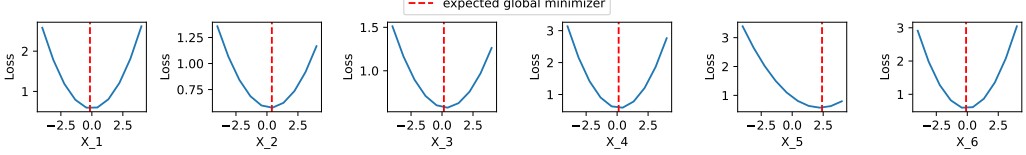

Figure 12: Loss function w.r.t. the first six parameters after reparameterization.

# E  FORWARD PASS AS MULTI-OBJECTIVE OPTIMIZATION

To demonstrate the equivalence between the forward pass and preconditioned gradient descent, we aim to express the iterative definition of LSA as an update of weight vectors, drawing inspiration from Ahn et al. (2023). However, unlike their approach, our proof diverges because the update formula for LSA cannot be simplified due to the presence of nonzero entries in $b_l$.

**Proposition 6 (*Proposition 3 restated*).** *Consider the $L$-layer transformer parameterzed by $b_l$, $A_l = \begin{bmatrix} -\bar{A}_l \\ 0_{1 \times d} \end{bmatrix}$ where $b_l \in \mathbb{R}^{d+1}$, $\bar{A}_l \in \mathbb{R}^{d \times d}$ for $l \in [L]$. Let $y_{n+1}^{(l)}$ be the bottom-right entry of the $l$th layer output. Then $y_{n+1}^{(l)} = \langle w_l^{gd}, x_{n+1} \rangle$ where $w_l^{gd}$ is iteratively defined as follows: $w_0^{gd} = 0$ and*

$$w_{l+1}^{gd} = w_l^{gd} - b_l^\top \nabla R(\theta) \bar{A}_l \quad \text{where } R(w) = \frac{1}{n} \sum_{i=1}^n \begin{bmatrix} -x_i \otimes \langle w, x_i \rangle \\ (\langle w, x_i \rangle - y_{n+1})^2 \end{bmatrix}$$

*Proof.* Let $y_i^{(k)}$ denote the $(d+1)$-$i$ entry of the embdding $Z_k$ and $x_i^{(k)}$ is the first $d$ entries of the $i$th column in $Z_k$. Since the first $d$ rows of $P$ is zero, the first $d$ rows of $Z_k$ is the same as $Z_0$. Therefore $x_i^{(k)} = x_i^{(0)} = x_i$, $\forall i \in [n+1]$.

We define a mapping to represent applying $k$ transformer layers to the bottom right entry of an embedding matrix $Z_0$ with $[Z_0]_{d+1,n+1} = y$: $g(x, y, k) : \mathbb{R}^d \times \mathbb{R} \times \mathbb{Z} \to \mathbb{R}$. When $x = x_{n+1}, y = y_{n+1}^{(0)} = 0$, $g(x, y, k) = g(x, 0, k) = y_{n+1}^{(k)}$. We establish two claims for $g(x, y, k)$ when $x = x_{n+1}$.

**Claim 1:** $g(x, y, k) = g(x, 0, k) + y$. The equation implies that applying the transfomer $k$ times on $Z_0$ with $[Z_0]_{d+1,n+1} = y$ is equivalent to applying the transformer $k$ times on $Z_0'$ with $[Z_0']_{d+1,n+1} = 0$ and then add the resulting bottom-right entry with $y$.

By definition of LSA, the iterative definition of $y_i^{(k)}$ ($i \in [n+1]$) is given by:

$$y_i^{(k+1)} = y_i^{(k)} - b_k^\top \underbrace{\frac{1}{n} \sum_{j=1}^n \begin{bmatrix} x_j x_j^\top & y_j^{(k)} x_j \\ y_j^{(k)} x_j^\top & y_j^{(k)2} \end{bmatrix}}_{:=\mathsf{G}^{(k)}} A_k x_i \tag{32}$$

Since $y_i^{(k)}$ is independent of $y_{n+1}^{(k')}$ for any $k'$, and $y_{n+1}^{(k)}$ depends on $y_{n+1}^{(k)}$ additively, one can show inductively that $g(x, y, k)$ and $g(x, 0, k)$ always differ by $y$, i.e., $g(x, y, k) = g(x, 0, k) + y$.

**Claim 2:** $g(x, 0, k)$ **is linear in** $x$. We prove the claim inductively. When $k = 0$, $g(x, 0, 0) = y_{n+1}^{(0)} - b_k^\top \mathsf{G}^{(k)} A_k x_{n+1}$ is linear in $x = x_{n+1}$. For $k \geq 0$, suppose $g(x, 0, k)$ is linear in $x$, then $g(x, 0, k+1) = y_{n+1}^{(k+1)} = y_{n+1}^{(k)} - b_k^\top \mathsf{G}^{(k)} A_k x_{n+1} = g(x, 0, k) - b_k^\top \mathsf{G}^{(k)} A_k x_{n+1}$. The first term $g(x, 0, k)$ is linear in $y$. The term $y_j^{(k)}$ with $j \neq n+1$ does not depend on $x_{n+1}$ according to equation 32. Hence $b_k^\top \mathsf{G}^{(k)} A_k x_{n+1}$ is also linear in $x_{n+1}$.

Combining the two claims, we have

$$g(x, y, k) = g(x, 0, k) + y = \langle \theta_k, x \rangle + y \tag{33}$$

for some $\theta_k \in \mathbb{R}^d$ with $\theta_0 = 0$. One can copy the values in the $i$th column to the $n+1$th column and adopt the previous arguments to show that $g(x_i, y_i, k) = \langle \theta_k, x_i \rangle + y_i$. By substituting $y_i = \langle \theta_k, x_i \rangle + y_i$ into equation 32, we have

$$y_{n+1}^{(k+1)} = y_{n+1}^{(k)} - \frac{1}{n} \sum_{j=1}^n b_k^\top \begin{bmatrix} x_j x_j^\top & (y_j + \theta_k^\top x_j) x_j \\ (y_j + \theta_k^\top x_j) x_j^\top & (y_j + \theta_k^\top x_j)^2 \end{bmatrix} A_k x_{n+1}$$

$$\Rightarrow \langle \theta_{k+1}, x_{n+1} \rangle = \langle \theta_k, x_{n+1} \rangle - \frac{1}{n} \sum_{j=1}^n b_k^\top \begin{bmatrix} x_j x_j^\top & (y_j + \theta_k^\top x_j) x_j \\ (y_j + \theta_k^\top x_j) x_j^\top & (y_j + \theta_k^\top x_j)^2 \end{bmatrix} A_k x_{n+1}$$

Since the above equation holds for any $x_{n+1}$, we obtain

$$\theta_{k+1} = \theta_k - \frac{1}{n} \sum_{j=1}^n b_k^\top \begin{bmatrix} x_j x_j^\top & (y_j + \theta_k^\top x_j) x_j \\ (y_j + \theta_k^\top x_j) x_j^\top & (y_j + \theta_k^\top x_j)^2 \end{bmatrix} A_k. \tag{34}$$

Here we interpret the RHS via the expectation over $y_j$. This corresponds to having multiple training prompts with the same $x_i, p_{01}, p_{11}$ but distinct $y_j$.

$$\theta_{k+1} = \theta_k - \frac{1}{n} \sum_{j=1}^n b_k^\top \begin{bmatrix} x_j x_j^\top & (\mathbb{E}[y_j \mid p_{01}, p_{11}, x_j] + \theta_k^\top x_j) x_j \\ (\mathbb{E}[y_j \mid p_{01}, p_{11}, x_j] + \theta_k^\top x_j) x_j^\top & (y_j + \theta_k^\top x_j)^2 \end{bmatrix} A_k.$$

Since the last row of $A_k$ is zero,

$$\theta_{k+1} = \theta_k - b_k^\top \underbrace{\frac{1}{n} \sum_{j=1}^n \begin{bmatrix} x_j x_j^\top \\ (p_{01} + (p_{11} - p_{01}) x_j + \theta_k^\top x_j) x_j^\top \end{bmatrix}}_{\bar{G} \in \mathbb{R}^{(d+1) \times d}} \bar{A}_k.$$

We treat $b_k, \bar{A}_k$ as the preconditioner. Let

$$R(\theta) = \frac{1}{n} \sum_{j=1}^{n} \begin{bmatrix} x_j[1]\theta^\top x_j \\ x_j[2]\theta^\top x_j \\ \vdots \\ x_j[d]\theta^\top x_j \\ (\theta^\top x_j + y_j)^2 \end{bmatrix} = \begin{bmatrix} \theta^\top \left( \frac{1}{n} \sum_{j=1}^{n} x_j[1]x_j \right) \\ \theta^\top \left( \frac{1}{n} \sum_{j=1}^{n} x_j[2]x_j \right) \\ \vdots \\ \theta^\top \left( \frac{1}{n} \sum_{j=1}^{n} x_j[d]x_j \right) \\ \frac{1}{n} \sum_{j=1}^{n} (\theta^\top x_j + y_j)^2 \end{bmatrix} = \frac{1}{n} \sum_{i=1}^{n} \begin{bmatrix} -x_i \otimes \langle w, x_i \rangle \\ (\langle w, x_i \rangle - y_{n+1})^2 \end{bmatrix}.$$

Then $\nabla R(\theta) = \bar{G}$ and

$$\theta_{k+1} = \theta_k - b_k^\top \nabla R(\theta) \bar{A}_k.$$

By letting $w_k^{gd} = -\theta_k$, we obtain the desired result. $\qquad \square$

## F   ADDITIONAL RELATED WORK

**Time series prediction.**    Time series prediction problems can be categorized into transductive and inductive setups. In the transductive case, a model (e.g., recurrent neural networks, neural ordinary differential equations) is trained on the initial portion of a new sequence and then used to predict future time steps for that same sequence. The next-token prediction for first-order binary Markov chains has been addressed in this context (Makkuva et al., 2024), demonstrating that transformers can effectively learn to output the transition probabilities of the input sequence. However, the global minimum in their case has trivial attention parameters, indicating that the absence of attention can still yield the desired performance. In contrast, our study requires that attention parameters be non-zero to reach the global minimum.

On the other hand, in inductive scenarios (Kipf et al., 2018; Huang et al., 2021; Li et al., 2020), a model is trained on multiple time series derived from the same dynamics. During inference, the trained model uses partial observations from an unseen time series sharing the same dynamics to predict future time steps without the need for fine-tuning. In this case, the learned model captures the dynamics from the observational window and makes predictions accordingly. However, ICL extends beyond this framework by addressing a higher-level problem that involves learning across various dynamical systems with different parameter settings, such as transition probabilities in the case of Markov chains. In this case, the trained transformer infers the unseen dynamical system from the in-context samples and makes predictions for the query sequence.

