# OpenReview forum: "Global Optimality of In-context Markovian Dynamics Learning"
_ICLR.cc/2025/Conference — Submitted to ICLR 2025_

### Official Review · Reviewer_mHik · 2024-10-23

**Soundness:** 2
**Presentation:** 3
**Contribution:** 1
**Rating:** 3
**Confidence:** 4

**Summary:**

This paper investigates how linear self-attention (LSA) transformers learn in-context from Markov sequence data. For single-layer, single-head LSA and length 2 binary random Markov chains, the global minimum of the loss landscape is derived in closed form. In particular, the minimum is shown to have a denser structure compared to the minimum for ICL of linear regression tasks. A bound for the loss of the global minimum in the case of arbitrary-length Markov chains is also given. For multi-layer transformers, the forward pass is interpreted as preconditioned gradient descent minimizing multiple objectives (involving both squared loss and linear maps). Moreover, the ability of transformer layers to learn Markov chains in context is verified by numerical experiments with GPT-2 blocks.

**Strengths:**

The studied ICL task of next-token prediction for Markov chains is closely related to N-gram models of language and is a welcome departure from the usual linear regression setting, which has perhaps been overanalyzed despite its considerable simplifications. The paper is overall well-organized and easy to follow. The theoretical results are backed by numerical experiments on practical architectures.

**Weaknesses:**

Overall, the theoretical contributions feel disjointed, too specific/abstract or not wholly justified, and do not give much insight into the structure of the problem. My reasoning is as follows.
* Propositions 1,2 are very specific computations and do not seem to have any bearing beyond the length 2 binary chain setting. Simply writing down a closed-form expression of the global optimum in the particular 2x2 case does not give any insight into the general problem of ICL of Markov chains. Since many ICL theory papers including the current work assume a very stylized setup (e.g. simplified architecture) I believe it is important to have takeaways that can potentially be generalized. The paper makes an attempt to address this by contrasting the denser structure of the minimum with the sparser structure for ICL of linear regression observed by Ahn et al. (2023), which is a valid point. However, this is not particularly surprising as we have no reason to believe the attention mechanism should become sparse a priori; it is the latter result that is unexpected.
* In a similar vein, the paper claims to analyze the loss landscape of ICL but does not have any results beyond characterizing the global minimum. For example, Zhang et al. (2023) show a global PL condition for ICL of linear regression and are able to conclude that gradient descent finds the global optimum. Can the techniques of this work be extended to obtain implications for optimization dynamics?
* In contrast to the first point, Theorem 1 seems to be too trivial and abstract to be of use. The theorem basically says that if $f=g\circ h$ then $\min f\ge\min g$ and $\min f\le f(x')$ for $x'$ such that $h(x^*)$ close to the minimizer of $g$, which is trivially true in any situation. This by itself is not an issue as it still suggests a potential recipe for obtaining concrete bounds. However, this result is subsequently used only in Example 2 in the Appendix, which currently looks to be incomplete and needs to be fixed: the computations for the projection are missing and no loss bounds are actually obtained. Also, Theorem 1 seems to never use the strict convexity of $\tilde{f}$.
* Even if loss bounds can be obtained using Theorem 1, it is unclear how they relate to Propositions 1,2 which studied the structure of the optimum, as loss values were not evaluated at the optimum or otherwise discussed previously. This needs to be clarified in the paper.
* Proposition 3 assumes a modified sparsity constraint (12) by zeroing out the last row of $Q$, which is different from the actual dense structure observed in Propositions 1,2 for the 2x2 case and similar to the optimum for ICL with linear regression. This is not justified in the paper except to align the dimensions to obtain the construction $R$, and seems to only exist for the sake of deriving another "forward pass = preconditioned GD" result.


Ahn et al. Transformers learn to implement preconditioned gradient descent for in-context learning. NeurIPS 2023.

Zhang et al. Trained transformers learn linear models in context. 2023.

**Questions:**

* Besides the derivation of the optimum, what other properties of the loss landscape can be derived, and can anything be said about the convergence properties of gradient descent on this landscape?
* Propositions 1,2 derive the optimal parameters, while Theorem 1 only gives bounds on the loss value at the optimum; how are these results connected?
* How is the modified sparsity constraint of Proposition 3 justified in view of the dense structure derived in Propositions 1,2?

---

> ### Author Response · Authors · 2024-11-24
>
> We appreciate your review and constructive feedback. We hope our response will be helpful for resolving the questions you raised.
>
> - **Restricted scope for Propositions 1 and 2:**
>     - **Generalizability**: While our Proposition 1, 2 focus on linear self-attention models applied to length-2 binary Markov chains, the underlying framework and theoretical results can be  generalized to more complex architectures and arbitrary-length Markov chains. Specifically, Lemma 4 and Theorem 1 provide a pathway for locating parameters that either exactly match or closely approximate the global minimum for Markov chains of any length. Moreover, Example 2 in the manuscript demonstrates that the denser global minimum structure we observed persists for longer chains, supporting the broader applicability of our findings. Additionally, the interpretation of the transformers as performing multi-level optimization (Proposition 3) extends beyond the binary case, providing deeper insights into the expressivity and capabilities of transformers.
>     - **Contribution and insights:** Our results provide a novel interpretation of ICL for Markov chains with sequence-level attention   in transformers, revealing how the global minimum structure shapes their forward pass differently compared to ICL for linear tasks [Ahn et al. NeurIPS'23]. Compared to prior work on ICL for structured data [Rajaraman et al., 2024; Nichani et al., ICML'24; Sander et al., ICML'24], our study shifts the focus from token-level attention to sequence-level attention, capturing relationships across multiple sequences. This approach offers a fresh perspective on how transformers handle dependencies between sequences in Markovian settings.
>     - **Potential application:** This framework has potential applications in tasks such as preferential alignment and dynamic recommendation systems, where modeling complex dependencies across sequences is critical. Therefore, the scope of the focused topic in this work is actually not narrower than existing works; it not only can be extended to non-binary values but also distinguishes itself by focusing on different aspects of the self-attention mechanism in Transformer models.
>
> - **Limited type of results and potential convergence analysis (Q1):** Existing research on ICL has predominantly focused on linear tasks with i.i.d. Gaussian-distributed input and task vectors, simplifying analysis but limiting applicability to real-world scenarios where data exhibits dependencies and context coherence. To bridge this gap, we leverage the Markovian data structure to capture these dependencies, extending ICL analysis to Markov chains with attention applied across entire sequences rather than within individual chains, as seen in related works [Rajaraman et al., 2024; Nichani et al., ICML'24].
> We  focus on characterizing the global minimum of the in-context loss landscape and studying the expressiveness of transformers parameterized by a certain set of parameters.
> Convergence analysis, on the other hand, stems from a distinct  perspective, focusing on the optimization dynamics and the pathways models take to approach these minima. While our study does not address this aspect, we acknowledge its importance and view it as an exciting direction for future research.
>
> - **Theorem 1 related questions:**
>     - **Lack of significance:** Theorem 1 establishes an error bound for arbitrary-length Markov chains, extending the applicability of our analysis beyond the specific cases addressed in Propositions 1 and 2.  Specifically, Theorem 1  provides a framework to identify analytic parameters in the reparameterzied space (leveraging Lemma 4) that can be projected to the LSA parameter space to either exactly match or closely approximate the global minimum of original loss, which shed light on the structure of the global minimum and the range of the global minima.
>     - **Lack of usage:**  We have included a numerical verification in the updated manuscript (Figure 10 in Appendix A.3). Specifically, we trained an LSA model via gradient descent and observed that the mean loss curve, averaged over multiple trials, lies well within the bounds. The dashed black and red lines, representing the expected lower and upper bounds derived in Theorem 1, closely align for length-2 Markov chains, as the global minimizer $X^*$ of $\tilde{f}$ can be exactly mapped to the transformer parameter space via $\phi(P, Q) = X^*$. For length-3 Markov chains, however, no such mapping exists, leading to a slightly looser bound with a difference of 0.12.
>     - **Strict convexity not used in Theorem 1:** Strict convexity (Lemma 3) is used in Lemma 4 to establish the existence and uniqueness of the optimal solution $X^*$ of the reparameterized objective function. This result enables us to use $X^*$ to obtain a set of transformer parameters that most closely align with $X^*.$

---

> > ### Author Response · Authors · 2024-11-24
> >
> > - **Unclear connection between loss bounds  and optimum structure  (Q2):**
> >     - **Obtained global minimum for reparameterization:** We would like to highlight that for arbitrary length, we are able to obtain the global minimum structure of the reparameterized objective function, as shown in Lemma 4. This result mirrors the case for Lemma 1 and 2, which serves as the foundation for Proposition 1 and 2, respectively.
> >     - **Possible non-existence of corresponding Transformer parameters:** Since the mapping from the transformer parameter space to the reparameterized space is not surjective, it is possible that we cannot find a solution in the transformer parameter space that can be mapped to the global minimum through Eq. (11) obtained in Lemma 4. However, the  analytic global minimum for the reparameterized objective still provides insights for the critical points and values for the ICL population loss.
> >
> > - **Lack of justification for the sparsity constraint in Proposition 3 (Q3)** The global optimum derived in Proposition 1 and 2 motivates the constraints in Eq. (12). By Proposition 1,2, we found that for Markovian data, the entries in the last row of the projection matrix $P$ needs to be nonzero. Although the last row of $A$ should also be nonzero, this makes it impossible to interpret the forward pass as the optimization of some other objective function due to the mismatch between the dimension of the in-context data vector $d$ and the number of nonzero rows $d+1$ in $A$. Thus, we assume the last row of $A$ is zero. This constraint extends prior work's understanding of the forward pass' expressivity, by considering more nonzero entries in $P.$
> >
> > **References**
> >
> > 1. Ahn, Kwangjun, et al. "Transformers learn to implement preconditioned gradient descent for in-context learning." Advances in Neural Information Processing Systems 36 (2023): 45614-45650.
> > 2. Rajaraman, Nived, et al. "Transformers on markov data: Constant depth suffices." arXiv preprint arXiv:2407.17686 (2024).
> > 3. Nichani, Eshaan, Alex Damian, and Jason D. Lee. "How Transformers Learn Causal Structure with Gradient Descent." Forty-first International Conference on Machine Learning (ICML)
> > 4. Sander, Michael Eli, Raja Giryes, Taiji Suzuki, Mathieu Blondel, and Gabriel Peyré. “How Do Transformers Perform In-Context Autoregressive Learning?” Forty-first International Conference on Machine Learning (ICML)

---

> ### Author Response · Authors · 2024-11-28
>
> Dear Reviewer mHik,
>
> We sincerely appreciate your valuable suggestions. We hope our response and revisions effectively address your feedback. Should there be any remaining questions or concerns, we would be happy to address them promptly.
>
> Thank you for your time and thorough review.
>
> Best regards,
>
> Authors

---

> ### Comment · Reviewer_mHik · 2024-11-29
>
> Thank you for the detailed response. I am satisfied with the authors' clarification of Theorem 1 with Figure 10, and their response to Q1, as optimization analysis seems very challenging in this setting. However, I still wish to discuss the first and last points on the main message of the paper.
>
> * **Restricted scope:** While I see that Lemma 4 handles longer chains, I do not understand how the theoretical results (explicit computations of minima) can be generalized to more complex architectures. According to the authors' response, the main takeaway seems to be that the global optimum has a denser structure compared to ICL of linear regression. However, as I mentioned, **this is not surprising as we have no reason to believe the attention mechanism should become sparse a priori**; it is the latter result that is unexpected. For another example, in the softmax case, an interesting result could be to discover that scores tend to concentrate on specific tokens, or conversely that certain scores tend to become uniform, which are intuitive takeaways that can be utilized when studying ICL of LLMs in practice. Moreover, optimization and convergence guarantees tell us that certain optima can actually be found through gradient descent in practice. How do the computations in Example 2 or Lemma 4, say, help us understand ICL in practice? (One answer could be the forward pass interpretation in Proposition 3, however see the comment below.)
> * **Lack of justification for the sparsity constraint:** "Although the last row of $A$ should also be nonzero, this makes it impossible to interpret the forward pass as the optimization of some other objective function... Thus, we assume the last row of $A$ is zero." **This is precisely my issue:** as I mentioned, there is no real justification besides to be able to derive a "forward pass = optimization" result, and directly contradicts the "denser structure" takeaway of the previous results. Can this be at least justified through e.g. experiments showing that the last row of $A$ is close to zero?

---

> > ### Author Response · Authors · 2024-12-01
> >
> > We appreciate your constructive questions and are encouraged to see your recognition of our response to Theorem 1 and the challenge of our setting.
> >
> > **Restricted scope:**
> > - **Novel insights from problem setup (with attention map results):** Our work explores ICL for Markov chains using linear self-attention models through the perspective of loss landscape. Compared to ICL for linear regression [Ahn et al. NeurIPS'23], the in-context example in our case have a distinct dependency within the in-context input vectors and the relationship between the input-output pairs. In contrast to concurrent studies on ICL for sequences [Rajaraman et al. 2024, Nichani et al. ICML'24, Sander et al. ICML'24], our setup applies the attention mechanism across entire sequences rather than individual tokens. The learned attention maps extracts the attention structure among the sequences instead of reflecting the local dependence between two successive states. In Appendix A.2, we compare attention maps across three ICL tasks: sequence-level Markov chains (this work), token-level Markov chains, and linear regression. Using GPT-2 models trained with both MSE and cross-entropy losses, we find that sequence-level attention lies between the patterns observed in token-level and linear tasks.
> >     - As shown in Figure 9, similar to the linear case, the attention map under this work's setting is mostly evenly distributed, with stronger intensity along the diagonal compared to other regions. Additionally, for ICL for sequences tasks, some transformer layers exhibit columnwise sparsity. Moreover, in our sequence-level attention setup, the sub-diagonal entries are less prominent, reflecting differences in the nature of the relationships modeled. In contrast, token-level attention setups exhibit more emphasis along the sub-diagonal, consistent with the causal structure of first-order Markov chains.
> > - **Insight from Proposition 1,2:** Regarding the insight of the global minimum results, we provide and prove a new interpretation of the forward pass, relaxing the assumption on the last row of $A$. The forward interpretation no longer relies on the assumption that the last row of $A$ being zero and therefore respects the global minimum structure derived in Proposition 1,2. Please refer to the next point for further details.
> >
> >
> > **Lack of justification for the sparsity constraint:**
> > In our previous discussion, we stated that it would be impossible to interpret the forward pass as optimizing another objective function without assuming the last row of $A$ is zero, which was made under the framework and assumptions adopted at that stage of our analysis.  We now provide an interpretation of the forward pass without assuming the last row of $A$ is zero. The proof reveals that the forward pass is equivalent to a multi-objective optimization problem with two groups of objectives $R_1(w)$ and $R_2(w)$, preconditioned by transformer parameters $b,A$. The objectives are defined as follows:
> > $$
> > R_1(w ) = \frac{1}{n}\sum_{j=1}^{n} \begin{bmatrix}-   \langle w ,x_j \rangle x_j \\\\
> > 0.5(y_{j}-\langle w ,x_{j}\rangle )^2 \end{bmatrix}
> > $$
> > $$
> > R_2(w) =   \frac{1}{n}\sum_{j=1}^n
> > \begin{cases}
> > \begin{bmatrix}
> >  (-w_{d}(y_j-\langle w_{:d-1},x_{j,:d-1}\rangle)+\frac{w_{d}^2}{2} x_{j,d}) x_j \\\\
> > \frac{1}{3x_{j,d}}(y_j - w^{\top} x_j)^3
> > \end{bmatrix}\quad \text{if $x_{j,d}\neq 0$}\\\\
> > \begin{bmatrix}
> >  (-w_{d}(y_j-\langle w_{:d-1},x_{j,:d-1}\rangle)+\frac{w_{d}^2}{2} x_{j,d}) x_j \\\\
> > -(y_j -\langle w_{:d-1},x_{j,:d-1}\rangle)^2 w_d
> > \end{bmatrix}\quad \text{if $x_{j,d}=0$}
> > \end{cases}
> > $$
> > $R_1(w)$ captures the interaction between individual states of $x_j$ and their contributions to the model’s prediction, aligning it with the target $y_j$. $R_2(w)$ focuses on the last entry of $w$ and introduces quadratic and cubic terms that adapt dynamically based on whether $x_{j,d}$ is zero.
> > The statement and proof are provided in the following anonymous link generated by Adobe: [https://acrobat.adobe.com/id/urn:aaid:sc:US:51186150-37d5-4ac1-8cb5-f28a1d6fdb6d](https://acrobat.adobe.com/id/urn:aaid:sc:US:51186150-37d5-4ac1-8cb5-f28a1d6fdb6d).

---

> ### Comment · Reviewer_mHik · 2024-12-01
>
> Thank you for updating Proposition 3. It is interesting to see that the dense case can also be expressed as preconditioned GD, although it is difficult to decipher what the new objective term means.
>
> While I agree that the problem setup can provide a new interpretation of ICL for Markov sequences, my comment on the lack of generalizability of the actual theoretical results still stands. Due to the efforts of the authors which clarified some of my concerns, I would be willing to increase my score to 4 if it were possible, nonetheless I believe the paper would still benefit from an additional round of revision.

---

> > ### Author Response · Authors · 2024-12-01
> >
> > We are encouraged to hear your willingness to reevaluate our work, and we sincerely thank you for recognizing the novelty of our problem setup. Your constructive suggestions have been invaluable in helping us improve our manuscript.

---

### Official Review · Reviewer_LZq8 · 2024-11-01

**Soundness:** 3
**Presentation:** 2
**Contribution:** 2
**Rating:** 5
**Confidence:** 3

**Summary:**

This work studies In context learning (ICL) of a Linear Self Attention (LSA) Model. In particular, it considers task distributions, where each task is represented by a Markov chain with binary states and the goal is to predict the last state of samples from the Markov chain. The authors address the problem of minimising the ICL loss in Expectation over the task distribution and present an explicit expression for the minimiser in the case of length-2 Markov chains (where for each tasks, the initial states can also not be i.i.d.). For the case of a Markov chain of arbitrary length, they provide lower and upper bounds for the minimum which depend on the minimiser of a strictly convex function, which is the expected loss under a transformation of the parameter of the LSA Model. Furthermore, they show that an LSA model with $l$ layers performs $l$ steps of preconditioned multi-objective gradient descent steps where in addition to the in-context objective it considers other linear objectives. The authors also provide experiments with both LSA and transformer models.

**Strengths:**

1. The theoretical results seem novel. As the author point out, most of the literature has focused on deterministic input-output mappings, while they consider a specific probabilistic mapping (given by a Markov chain with binary values).
2. The analysis seems sound: I checked mainly the proof of the first  two propositions.

**Weaknesses:**

1. Clarity (see questions section).
2. Limited scope of the theoretical results. The setup with binary values is quite restrictive. Moreover, Theorem 1 for arbitrary lengths Markov chain only provides bounds to the global minimum whose tightness is unclear.

**Questions:**

Questions:
1. Can the authors explain why the  lower/upper bounds provided in Theorem 1 are tight?
2. I found the statement in proposition 3 (and the whole section 3.2) unclear. Is gradient descent the minimiser of the expected loss? Under what assumptions does the proposition hold?
3. Concerning proposition 3, can the authors provide an intuition or show some empirical consequences of having the other linear objectives?
4. I found the use of the dense/sparse when referring to the global Minimiser in the Markovian and non-Markovian settings unclear before looking at Figure (1). Can you clarify what this means in the introduction? Also, Can you provide some intuition on why the global minima has more terms when applied to markovian data?
5. Can you explain how to pass from Lemma 1 to Proposition 4, even a short paragraph as a proof of proposition 4 would suffice.

Comments:
1. The task column in Table 1 labels this work as addressing next-token prediction. This terminology is also used in other parts of the paper. I found this misleading because the task is still formulated as a supervised regression task that differs from previous work by having a random mapping from input to outputs. Meanwhile, for next token prediction I would expect the examples to be necessarily linked temporally as e.g. In [1] (which might be added as a reference). I suggest the authors to change this terminology and possibly remove that column.
2. Line 198, LSA is not directly linked to eq 6, which might make reading hard for people unfamiliar with such equation. Consider changing Attn to LSA in eq. 6 and following equations (also not to use the same symbol of Softmax attention in e. (5).).
3. Square bracket notation, used from equation (11) onwards for indices of vectors, is rather strange and hard to read. I suggest the authors to change it, maybe using a double index.
4. First Experimental section could benefit from an introductory paragraph giving an outline.

Minor:
- Propositions should probably be theorems since they are part of the main contributions of the work.
- Line 135: sampled from usually is followed by a distribution, not by the input and output spaces. Consider using “belong to” instead.
- Line 229: (1) Data -> (a) Data.
- In the definition of c_1 of Proposition 2, j should be replaced by n+1.
- $|\mathcal{S}|$ in evaluation metrics is undefined. Are these still binary values?

[1] Sander, Michael Eli, et al. "How do Transformers Perform In-Context Autoregressive Learning?." Forty-first International Conference on Machine Learning.

---

> ### Author Response · Authors · 2024-11-24
>
> We appreciate your thoughtful feedback and are encouraged by your recognition of the novelty of our work in advancing the understanding of ICL for stochastic mappings.
>
> - **W1. clarity:** We apologize for any confusion and hope our response to the specific questions below could address your concerns.
>
> - **W2. Limited scopes of results:** We would like to highlight the unique challenge of this work, its generalizability and broader application.
>     - **Novelty**:  Existing studies on transformers' in-context learning have primarily focused on linear tasks with i.i.d. Gaussian-distributed input and task vectors. While this assumption simplifies the analysis and provides initial insights into the attention mechanism, it is less practical for real-world scenarios where data often exhibits dependencies and context coherence. To address this gap, we leverage the Markovian data structure, which captures such dependencies and provides a more realistic framework for analyzing self-attention mechanisms. Specifically, we extend the study of in-context learning to Markov chains, retaining the prompt construction where attention weights are applied across Markovian sequences. This setup enables us to characterize the behavior of transformers in handling stochastic dependencies that go beyond the simplified i.i.d. setting.
>     - **Extensibility**: While our characterization of the global minima focuses on binary Markov chains, our theoretical framework can be extended to Markov chains with state spaces of arbitrary size, allowing for broader applicability and further exploration of complex probabilistic mappings. Additionally, the interpretation of the transformers as performing multi-level optimization (Proposition 3) extends beyond the binary case, providing deeper insights into the expressivity and capabilities of transformers.
>     - **Theorem 1 implication:**  Theorem 1  provides a framework to identify analytic parameters in the reparameterzied space (leveraging Lemma 4) that can be projected to the LSA parameter space to either exactly match or closely approximate the global minimum of original loss, which shed light on the structure of the global minimum and the range of the global minima. We numerically verified the tightness of the bound in Appendix B, as detailed below.
>     - **Potential application:** Our study of ICL for Markov chains provides a framework for understanding how transformers aggregate sequence-level dependencies, capturing relationships that extend beyond token-level associations. This framework has potential applications in tasks such as preferential alignment and dynamic recommendation systems, where modeling complex dependencies across sequences is critical. Therefore, the scope of the focused topic in this work is actually not narrower than existing works; it not only can be extended to non-binary values but also distinguishes itself by focusing on different aspects of the self-attention mechanism in Transformer models.
>
> - **Q1. Tightness of the bound** We have included new numerical verification in the updated manuscript (Figure 10 in Appendix A.3). Specifically, we trained an LSA model via gradient descent and observed that the mean loss curve, averaged over multiple trials, lies well within the bounds. The dashed black and red lines, representing the expected lower and upper bounds derived in Theorem 1, closely align for length-2 Markov chains, as the global minimizer $X$* of $\tilde{f}$ can be exactly mapped to the transformer parameter space via $\phi(P, Q) = X^{*}$. For length-3 Markov chains, however, no such mapping exists, leading to a slightly looser bound with a difference of 0.12.
>
> - **Q2. Lack of clarity in Proposition 3 and Section 3.2**  Proposition 3 does not involve taking the expectation of the objective; it holds for an arbitrary instance of the prompt. The only assumption required is that the structure of the global minimizer satisfies the sparsity constraint specified in Eq. (12). We appreciate your feedback and have revised the section accordingly.  Please see the updated manuscript.
>
> - **Q3. Insight of Proposition 3** Proposition 3 highlights a key insight into the mechanism of the transformer for ICL in Markov chains. When interpreting the forward pass of transformers, we aim to identify objective functions whose gradients produce the in-context data terms $\{x_i,y_i\}_{i=1}^{n}$, treating the transformer parameters as preconditioned matrices. In the linear case, only a single entry in the optimal projection matrix is nonzero, effectively canceling out the linear objectives and leaving only the square loss. In contrast, for the Markovian case, all entries in the optimal projection matrix of the LSA influencing the output are nonzero, resulting in the retention of linear objectives. This mechanism allows the forward pass of the transformer to be interpreted as minimizing a multi-objective loss, balancing the square loss and linear objectives.

---

> > ### Author Response · Authors · 2024-11-24
> >
> > - **Q4. Clarification of dense/sparse structure of global minimizers**
> > [Ahn et al., NeurIPS'23] identified a sparse structure in the globally optimal LSA parameters for linear tasks, with   a single   entry in the projection matrix and   the top-left $d \times d$ block of the attention matrix nonzero. In contrast, for sequentially dependent in-context samples, achieving the global minimum of the loss requires additional nonzero parameters in the Transformer, reflecting the increased complexity of capturing temporal dependencies. Specifically,
> >     - To evaluate the global minimum of LSA, we reframed the problem using a reparameterized objective, where the gradient depends on the in-context data $[x_i; y_i]$. In the linear case, the zero-mean distributions of $x_i$ and $w$, combined with the  relationship between $x_i$ and $y_i$ being linear ($y_i=w^{\top}x$) without bias, result in the data matrices in the gradient and the reparameterized global minimum having  mostly zero entries. When mapped back to the transformer parameter space, this yields a sparse structure in the global minimizer.
> >     - In the Markovian case, however, the sequential dependence breaks the zero-mean property of the in-context samples. Further, the relationship between in-context input and output  has an additional ``bias'' term related to the transition probability, after taking expectation. These lead  to a denser global minimizer in the reparameterized objective, which influences the global minima of the original loss landscape.
> >
> > - **Q5. Transition fron Lemma 1 to Proposition 4**  The mapping between the reparameterized and original parameters for $d=1$ is given as: $X_1 = a_1b_1$, $X_2 = a_1b_2 + a_2b_1$, $X_3 = a_2b_2$. (This correspondence is also detailed in Section D.1 for reference.) To proceed, we set $b_1 = 1$, which yields $a_1 = X_1$. Substituting this into the second equation gives: $a_1b_2 + a_2 = X_1b_2 + a_2 = X_2$. Rearranging for $a_2$, we obtain: $a_2 = X_2 - X_1b_2$. Substituting $a_2$ into the third equation, $X_3 = a_2b_2$, we solve for $b_2$, and the expression for $a_2$ follows directly.
> >
> > - **Comment 1. Potential confusion regarding ``next-token prediction''** We apologize for the potential confusion. We agree that this terminology could be misleading, as our formulation is based on a supervised regression objective. However, we would like to clarify that while the analyzed objective is regression, the in-context inputs and outputs are sequentially dependent, generated from Markovian dynamics. Moreover, the specific output of the transformer corresponds to the next token in the query sequence. To address this concern, we have removed the column in Table 1 to reduce ambiguity, and compared with  [Sander et al. ICML'24] for clarity.
> >
> > - **Comment 2-4.**  Thank you for your careful reading and detailed feedback. We have incorporated the suggested changes, highlighted in blue text, into the updated manuscript.
> >     - **C2.** We clarified our notation regarding LSA in the updated manuscript. To differentiate Eq. (6) (linear attention)  with Eq. (5) (nonlinear attention), we added a superscript in Eq. (6) to indicate that the attention mechanism is linear.
> >     - **C3.** We replaced square brackets with a double subscript for matrix indexing.
> >     - **C4.** We added an introductory paragraph that outlines the experiments in Section 3.3.
> >
> > - **Minor issues.**   We have addressed these points in the updated manuscript. We kept the term "Proposition" for the first two results to ensure unambiguous referencing in the rebuttal, and we will revise the naming in the final version. As for the undefined $|S|$, we replaced ${0,\ldots,|S|-1}$ with ${0,1}$ to explicitly indicate that the state space for empirical evaluations in Section 4 is binary.
> >
> > **References**
> >
> > 1. Ahn, Kwangjun, et al. "Transformers learn to implement preconditioned gradient descent for in-context learning." Advances in Neural Information Processing Systems 36 (2023): 45614-45650.
> > 2. Sander, Michael Eli, Raja Giryes, Taiji Suzuki, Mathieu Blondel, and Gabriel Peyré. “How Do Transformers Perform In-Context Autoregressive Learning?” Forty-first International Conference on Machine Learning (ICML)

---

> ### Author Response · Authors · 2024-11-28
>
> Dear Reviewer LZq8,
>
> We are grateful for your constructive suggestions. In response, we have revised our submission and incorporated additional results to address your comments. Please don’t hesitate to let us know if there are any remaining questions or concerns.
>
> Thank you for your time and thoughtful review.
>
> Best regards,
>
> Authors

---

> > ### Comment · Reviewer_LZq8 · 2024-12-01
> >
> > Sorry for the late reply. I decided to delay my response also because I shared the concerns of reviewer mHik about the theoretical results and was monitoring your discussion with them.
> > I appreciate the update to the manuscript that incorporated many changes and improved the quality. I still think there are remaining issues which prevent me from changing the score.
> >
> > **Proposition 3.** I still do not understand what insights does proposition 3 provide. The updated proposition introduces even more terms in the loss whose interpretation  is unclear. Regarding this point and the other proposition, I share a similar concern as reviewer mHik, which mentions how such a proposition still lacks either theoretical or empirical justification.
> >
> > **(Minor) Tightness of the bounds.** The empirical results reported in the appendix show the tightness of the lower bound in the case of length 3 Markov chain. It would be nice, since the results are for arbitrary Markov chain to show (maybe theoretically) how the bound can be tight for arbitrary length.

---

> > > ### Author Response · Authors · 2024-12-01
> > >
> > > Thank you for the careful reading and further questions.
> > >
> > > - **Interpretation of Proposition 3**
> > >     - The new multi-objective formulation incorporates linear, quadratic, and cubic terms. The linear terms emphasize correlations within the in-context input sequence $x_j$ ($j \in [n]$), while the quadratic and cubic terms connect both the in-context input and output $x_j, y_j$, aiming to align the linear prediction $\langle w^\top y_j \rangle$ with the target $y_j$.
> > >     - For ICL of linear tasks with sparse i.i.d. data, the linear and cubic objectives are absent. The presence of these objectives in the current setting is due to the Markovian nature of the data, which introduces conditional dependencies within the input vector, as well as the general data structure with cross-sequence correlations as suggested in Proposition 2.
> > >     - Markovian data creates dependence both within and across sequences, resulting in more complicated correlation structure within the input prompt. Consequently, more Transformer parameters are required to effectively model the in-context data.
> > > - **Error bound**
> > >     - As demonstrated by our numerical results (Figure 9, Appendix B), the error bound remains tight for Markov chains of lengths 2 and 3. Furthermore, the lower bound provides a critical metric for quantifying the performance limits of Transformers in a Markovian data setting. To the best of our knowledge, this study is still the first attempt to understand ICL for Markovian data with the Transformer attention mechanism applied between sequences.

---

### Official Review · Reviewer_TNfS · 2024-11-09

**Soundness:** 3
**Presentation:** 3
**Contribution:** 2
**Rating:** 5
**Confidence:** 3

**Summary:**

This paper advances the theoretical understanding of in-context learning (ICL) in transformers beyond linear regression with i.i.d. inputs, focusing instead on next-token prediction for Markov chains using linear self-attention (LSA). The authors derive a closed-form solution for the global minimizer of ICL loss for single-layer LSA on length-2 Markov chains, emphasizing the increased structural complexity caused by Markovian dependencies and establishing bounds for arbitrary-length chains. They demonstrate that multilayer LSA models structured like the global minimizer perform preconditioned gradient descent on a multi-objective function.

**Strengths:**

1. The paper addresses an important problem that holds significant relevance to the community.
2. The paper is well-written and clearly presented.

**Weaknesses:**

1. I believe the setting might be somewhat simplistic concerning the broader applicability of this work to the community.
2. Please see questions.

**Questions:**

1. I am unclear about the exact setting here. In in-context learning for Markov chains (see references a and b), the model essentially reduces to a counting-based estimator, ensuring that it estimates the next token probability for the new Markov chain. In this context, does optimizing equation 8 in the paper imply that the model can estimate the transition probability for any new Markov chain provided? I am unsure if this is true, and if it is, I would like to understand why.

2. Since works like a and b already exist, where the setting aligns more closely with real-world scenarios and the authors demonstrate that transformers can perform in-context learning on Markov chains, I am unsure how beneficial this work is to the larger community. That said, it would be helpful if the authors clarified how their work differs from references a and b, so that the differences are clearer.

3. In equation 10, what does $X_i^*$ represent? Is it simply the i'th entry of $X^*$?

4. In experiment 3.3, can we train the LSA as described in equation 6 and observe the attention maps along with $P$ and $Q$? Additionally, can we train a standard LSA using $W_k$, $W_q$, and $W_v$ as specified in equation 5 and then compare the three scenarios? This includes comparing the standard LSA with query, key, and value components, the LSA with $P$ and $Q$, and the LSA constructed by the authors (as described in equation 9).

5. Would it be possible to visualize the attention maps from the experiments in section 4 and compare them to those in references a and b? Although it may be challenging to match them exactly due to different constructions, this comparison could provide insights into whether training the model with MSE or cross-entropy leads to similar attention maps.

Paper Suggestion:

I think figure - 1 could be potentially improved. It took some time to exactly understand the message being conveyed.

References:

a) Rajaraman, Nived, et al. "Transformers on markov data: Constant depth suffices." arXiv preprint arXiv:2407.17686 (2024).

b) Nichani, Eshaan, Alex Damian, and Jason D. Lee. "How transformers learn causal structure with gradient descent." arXiv preprint arXiv:2402.14735 (2024).

---

> ### Author Response · Authors · 2024-11-24
>
> We thank you for the detailed and constructive review, and we are encouraged to hear your recognition that our studied problem is important. We hope our response would be helpful for resolving the questions you raised.
>
> - **W1. Simplistic setting** Although the considered setting is "simple" in terms of mimicking the Transformer model used in LLMs, it does not lose the key structure of the self-attention mechanism, which distinguishes it from traditional neural network architectures. This structure has been extensively explored in theoretical works aimed at understanding attention mechanisms across a wide range of in-context learning paradigms. For example, seminal works on the adoption of linear self-attention [Ahn et al. NeurIPS'23] demonstrate the effectiveness of this architecture for ICL of linear tasks. The intent of employing this model is to provide a simple yet clean analysis of ICL while retaining transformers’ essential functionalities. To the best of our knowledge, this work represents the first step toward understanding the attention mechanisms involved in extracting sentence-level relationships between prompts and queries. This serves as a complementary contribution to characterizing the expressiveness of Transformers for Markovian data.
>
> - **Q1. Setup clarification**
>     - **Sequence-level vs token-level attention**: Unlike the setups in references [Rajaraman et al. 2024, Nichani et al. ICML'24], where attention scores are computed among states within a single Markov chain, our approach calculates attention scores between entire sequences. This sequence-level attention mechanism introduces additional challenges, as it requires the model to infer relationships between aggregated representations rather than individual tokens. This task demands the abstraction of finer-grained details, moving beyond simpler patterns like the similarity between in-context samples and the query in linear tasks with i.i.d. data, or the direct correlation between successive tokens in local structures. Consequently, transformers trained under this scenario cannot guarantee exact transition probability estimation due to the increased complexity of the task. Overall, this setup enables a theoretical analysis of in-context learning under conditions that reflect more complex dependencies between sequences.
>
>
> - **Q2. Difference from related works** Compared to references [Rajaraman et al. 2024, Nichani et al. ICML'24], our work explores a complementary perspective by focusing on a controlled condition where transformers learn relationships between entire sequences rather than within individual Markov chains. We believe this distinction offers a unique angle on ICL for structured data, as it reveals transformer behavior in settings not fully covered by chain-specific transitions alone. Our theoretical interpretation reveals that the self-attention mechanism for sequence-level Markov data performs multi-objective optimization, setting it apart from existing theoretical results on i.i.d. data or token-level causal structure discovery. Furthermore, new numerical results in Appendix A.2 demonstrate the learned attention map in our setup extracts patterns among the sequences instead of reflecting the local dependence between two successive states as in the references.
>
>
> - **Q3. Notation of $X_i$**  Yes,  X_i^* denotes the $i$th entry of $X^{*} $. We've added this clarification immediately after introducing the notation.
>
> - **Q4. Comparing attention maps and weights for self-attention models** We have conducted additional experiments to compare the three variants of self-attention models, which are included in the updated manuscript (Appendix A.1).
>     - Figure 6 (Appendix A.1.1) compares the training loss curves for the three variants, showing that the sparse and dense versions of LSA perform equivalently in terms of convergence behavior, while the standard nonlinear self-attention model (NSA) demonstrates lower error.
>     - We visualized the attention scores at convergence in Figure 7 (Appendix A.1.2). The results reveal that the attention scores are generally evenly distributed, with a notable dominance along the diagonal, indicating a stronger emphasis on self-attention.
>     - Additionally, Figure 8 (Appendix A.1.3) shows the attention weights at convergence align with the patterns identified in Proposition 1, characterized by nonzero values along the last row of $P$ and the first column of $Q$.

---

> > ### Author Response · Authors · 2024-11-24
> >
> > - **Q5. Attention maps   for GPT-2**  We provide the simulation for attention maps of GPT-2 models to compare different problem setups and objective functions for ICL in Appendix A.2.
> >
> >     - Specifically, we compared attention maps across three in-context learning (ICL) tasks: (1) sequence-level attention for Markov chains (this work), (2) token-level attention for Markov chains [Rajaraman et al. 2024, Nichani et al. ICML'24], and (3) ICL for linear regression [Ahn et al. NeurIPS'23]. For each task, we trained a GPT-2 model with 3 layers and 2 attention heads using both MSE (for all three setups) and cross-entropy losses (for the first two setups) to investigate whether the problem setting and the loss function influences the learned attention patterns.
> >     - As shown in Figure 9, transformers in task (1) learn an attention map that lies between the other two settings. Similar to the linear case (task 3), the attention map is mostly evenly distributed, with stronger intensity along the diagonal compared to other regions. Additionally, for task (1) and (2), some transformer layers exhibit columnwise sparsity. Moreover, in our sequence-level attention setup, the sub-diagonal entries are less prominent, reflecting differences in the nature of the relationships modeled. In contrast, token-level attention setups (task 2) exhibit more emphasis along the sub-diagonal, consistent with the causal structure of first-order Markov chains. Despite these differences, the attention maps trained using MSE and cross-entropy loss within each setup display similar overall patterns, as both losses are designed to align predictions with the true labels.
> >
> > - **Suggestion for Figure 1.**  We have revised Figure 1 to illustrate the differences between our setup and those in related works. Specifically, we highlighted the distinction between the token-level attention mechanism used in the references and the sequence-level attention mechanism employed in our work.
> >
> > **References**
> >
> > 1. Ahn, Kwangjun, et al. "Transformers learn to implement preconditioned gradient descent for in-context learning." Advances in Neural Information Processing Systems 36 (2023): 45614-45650.
> > 2. Rajaraman, Nived, et al. "Transformers on markov data: Constant depth suffices." arXiv preprint arXiv:2407.17686 (2024).
> > 3. Nichani, Eshaan, Alex Damian, and Jason D. Lee. "How Transformers Learn Causal Structure with Gradient Descent." Forty-first International Conference on Machine Learning (ICML)

---

> ### Comment · Reviewer_TNfS · 2024-11-26
>
> Thank you for your response. I had a question regarding the setup. Could you clarify what you mean by the following:
>
> > Where attention scores are computed among states within a single Markov chain, our approach calculates attention scores between entire sequences.
>
> and
>
> > Our work explores a complementary perspective by focusing on a controlled condition where transformers learn relationships between entire sequences rather than within individual Markov chains. We believe this distinction offers a unique angle on ICL for structured data, as it reveals transformer behavior in settings not fully covered by chain-specific transitions alone.
>
> I'm a bit unclear on how this differs from the work of [Rajaraman et al. 2024, Nichani et al., ICML '24], where they also learn relationships across different sequences. Specifically, their approach involves computing the probability transition matrix using a central, common rule (a counting-based estimator), which then captures the relationships between different Markov chains. Could you elaborate on how your setup contrasts with theirs?

---

> > ### Author Response · Authors · 2024-11-27
> >
> > Thank you for the great question. We agree that the transformer functioning as a counting-based estimator [Rajaraman et al. 2024, Nichani et al., ICML'24] serves as a general rule for generating causal sequences, such as Markov chains. This counting-based rule can indeed be seen as capturing relationships between sequences, in the sense that it recovers the underlying mechanism of their generation. However, we characterize our setup as 'sequence-level attention,' as opposed to token-level attention, based on how the attention mechanism is applied to the given prompt. In prior works, the input prompt typically consists of a single Markov chain, with the attention mechanism learning relationships within that chain. In contrast, our input prompt includes multiple Markov chains, allowing the attention mechanism to capture relationships between the sequences in a prompt. To clarify this distinction further, we elaborate below using the attention mechanism formulas.
> >
> > - Token-Level
> >     - For simplicity, we omit the attention-weighted aggregation step and focus on the computation of attention scores  in [Rajaraman et al., 2024]. The relevant parameters include the key and query matrices, $\mathbf{W}_K$ and $\mathbf{W}_Q$. The input prompt consists of a Markov chain of length $n$ generated from some transition kernel. The embedding of the $j$-th token in the prompt is represented by  $\boldsymbol{x}_j$, and its relative positional encoding in its role as a key is denoted by $\boldsymbol{p}^K_j$. The attention score between query $i$ and key $j\in [n]$ is given as
> >
> >          $$\text{att}_{n,i} = \texttt{Softmax}  (\\{   \langle \mathbf{W}_K  (\boldsymbol{x}_j  + \boldsymbol{p}^{K}_j), \mathbf{W}_Q \boldsymbol{x}_j \rangle\\}) ,\ j\in[n]   $$
> >         Hence, the attention score determines the extent to which token $\boldsymbol{x}_i$ attends to token $\boldsymbol{x}_j$.
> >     - We consider the attention mechanism in [Nichani et al., ICML'24], parameterized by  $A\in \mathbb{R}^{d\times d}$. Let $\mathcal{S}$ denote the softmax function and $\text{MASK}$ denote a masking operator to ensure the attention is placed on previous tokens.  In the data embedding $h\in\mathbb{R}^{T \times d}$, each column represents a token in the input prompt.   The attention mechanism is defined as
> >          $$ attn(h; A) := \mathcal{S} (\text{MASK}(hAh^\top)) h   $$
> >         $\mathcal{S}(\text{MASK}(hA h^\top)) \in \mathbb{R}^{T \times T}$ computes the attention scores between tokens in the input causal sequence of length $T$. Specifically, $\mathcal{S}(\text{MASK}(hA h^\top))_{ij}$ ($i,j\in[T]$) represents how much focus token $i$ places on token $j$.
> >
> > - Sequence-Level
> >
> >     In our case, the attention mechanism is parameterized by matrices $P,Q\in \mathbb{R}^{d\times d}$. A prompt $Z\in \mathbb{R}^{(d+1)\times (n+1)}$ consists of $n+1$ Markov chains with length $d+1$. Note that the prompt construction differs from prior works in that each column of the prompt  $Z$ represents a Markov chain rather than a single token.   The attention is as follows
> >     $$
> >     Attn^{(\text{lin})}_{P,Q}(Z) = PZM(Z^{\top}QZ), M = \begin{bmatrix}I&0\\\\0&0\end{bmatrix}
> >     $$
> >     Here, the pairwise attention score is computed between each column of $Z$. In other words, the score determines the extent to which the $i$th sequence attends to the $j$th sequence.
> >
> > Therefore, there is a fundamental difference in how attention is used in this work compared to the two references, which can be attributed to differences in prompt construction. In our setup, different columns represent distinct sequences, each initialized with different token states drawn from a specific distribution. These sequences can model varying sentences or paragraphs, representing an ICL setting that, to the best of our knowledge, has not been explored in existing works.

---

> > > ### Comment · Reviewer_TNfS · 2024-12-03
> > >
> > > Thank you for the comments and the clarifications. I have gone over other responses as well basis which I am going to maintain my current score. Having said this, I think the setting is indeed interesting and would encourage the authors to resubmit the revised version to the next conference.

---

> > > > ### Author Response · Authors · 2024-12-03
> > > >
> > > > Thank you for your thoughtful feedback. Your comments and suggestions have been invaluable in improving our manuscript.

---

### Author Response · Authors · 2024-11-24

We thank the reviewers for your thoughtful feedback and hope that the responses below thoroughly address your questions and concerns. The changes have been incorporated into the updated manuscript, with revisions highlighted in blue text. Specifically, we have added supplemental experiments to  clarify the differences between our setup and related works in Appendix A and provide additional numerical verification for Theorem 1 in Appendix B. Below, we address the shared concerns raised by the reviewers and summarize the new numerical results.


- **Lack of novelty and limited scopes:** Reviewer TNfS raised concerns about the novelty of this work in comparison to related concurrent studies, while reviewers LZq8 and mHik expressed concerns regarding the limited scope of the setup. We appreciate these constructive comments and would like to take this opportunity to clarify the motivation, novelty, and technical challenges addressed. Additionally, we discuss a potential application of the proposed analysis mechanism to further emphasize its broader impact.
    - **Motivation**: Studies on transformers' in-context learning have revealed that the global minimum of the loss landscape allows the transformer's forward pass to be interpreted as preconditioned gradient descent. However, these findings are confined to linear tasks with i.i.d. Gaussian-distributed input and task vectors. Inspired by this framework, we seek to expand the understanding of ICL by exploring its application to Markov chains.
    - **Novelty**: Our work explores ICL for Markov chains using linear self-attention models through the perspective of loss landscape. Compared to ICL for linear regression [Ahn et al. NeurIPS'23], we reveal that the global minimum exhibits a denser structure, necessitating more nonzero parameters to achieve global optimality. In contrast to concurrent studies on ICL for sequences [Rajaraman et al. 2024, Nichani et al. ICML'24, Sander et al. ICML'24], our setup uniquely applies the attention mechanism across entire sequences rather than individual tokens. The learned attention maps extracts the attention structure among the sequences instead of reflecting the local dependence between two successive states, as shown in the new numerical results in Appendix A.
    - **Technical challenge**: The main challenge introduced by the Markovian structure arises from the inherent stochasticity in the relationship between inputs and labels, as well as the dynamic nature of the states, which are iteratively dependent on the initial state. Unlike the linear case, these complexities prevent a straightforward analytical determination of the expected values for input and output entries in the in-context learning process. We tackled these challenges by rigorously establishing the global minima in our theoretical analysis.
    - **Extensibility**: While we focus on binary Markov chains, our theoretical framework can be extended to Markov chains with state spaces of arbitrary size, allowing for broader applicability and further exploration of complex probabilistic mappings.
    - **Potential application:** Our study of ICL for Markov chains provides a framework for understanding how transformers aggregate sequence-level dependencies, capturing relationships that extend beyond token-level associations. This framework has potential applications in tasks such as preferential alignment and dynamic recommendation systems, where modeling complex dependencies across sequences is critical.  Therefore, the scope of the focused topic in this work is actually not narrower than existing works; it not only can be extended to non-binary values but also distinguishes itself by focusing on different aspects of the self-attention mechanism in Transformer models.


- **New numerical results:**
    - **Comparative analysis.** We conducted a comparative study focusing on two aspects: the model variations (Appendix A.1) and the in-context data modeling (Appendix A.2). Specifically, we analyzed the attention maps generated by three different variants of self-attention models and, separately, examined the attention maps across three distinct data setups while keeping the model fixed. Our findings reveal that the attention maps in some transformer layers are evenly distributed, similar to the linear case, and exhibit columnwise sparsity, akin to the token-level attention scenario. Notably, they lack the high-intensity values along the sub-diagonals associated with previous tokens, due to the differing applications of the attention mechanism.
     - **Tightness of bound.** Furthermore, we train the LSA model using gradient descent and numerically demonstrate that the converged loss closely adheres to the established bounds. The results are included in Appendix B.

---

> ### Author Response · Authors · 2024-11-24
>
> **References**
>
> 1. Ahn, Kwangjun, et al. "Transformers learn to implement preconditioned gradient descent for in-context learning." Advances in Neural Information Processing Systems 36 (2023): 45614-45650.
> 2. Rajaraman, Nived, et al. "Transformers on markov data: Constant depth suffices." arXiv preprint arXiv:2407.17686 (2024).
> 3. Nichani, Eshaan, Alex Damian, and Jason D. Lee. "How Transformers Learn Causal Structure with Gradient Descent." Forty-first International Conference on Machine Learning (ICML)
> 4. Sander, Michael Eli, Raja Giryes, Taiji Suzuki, Mathieu Blondel, and Gabriel Peyré. “How Do Transformers Perform In-Context Autoregressive Learning?” Forty-first International Conference on Machine Learning (ICML)

---

### Meta-Review · Area_Chair_HdnV · 2024-12-21

**Metareview:**

This paper theoretically investigates in-context learning ability of linear attention transform where the true distribution follows a Markov chain. The authors characterizes the global minimum and shows multilayer transformer can implement preconditioned gradient descent. Some numerical experiments are also provided to justify the theoretical results.

Although it is interesting to analyze in-context learning of transformers, this paper has several issues that should be fixed:
(i) The biggest issue is that the theoretical results presented in this paper do not really address the problem induced in the introduction. The statements are fragmented into several parts and they are not consistently connected into one story line. Moreover, the statements are not strong, that is, the problem settings are too restrictive or only vague bounds are given.
(ii) Relation to existing work is not thoroughly discussed. Some important literatures were missed in the first version.

For these reasons, I think this paper falls below the bar for acceptance.

**Additional Comments On Reviewer Discussion:**

The authors gave extensive responses to reviewers. However, those rebuttals did not convince the reviewers. The paper still requires substantial revision before publication.

---

### Decision · Program_Chairs · 2025-01-22

Reject